# Flood risk assessment for Indian sub-continental river basins

Urmin Vegad[1], Yadu Pokhrel[2], and Vimal Mishra[1,3*]

[1]Civil Engineering, Indian Institute of Technology (IIT) Gandhinagar, India

[2]Department of Civil and Environmental Engineering, Michigan State University, East Lansing, Michigan, USA

[3]Earth Sciences, Indian Institute of Technology (IIT) Gandhinagar, India

*Corresponding author: vmishra@iitgn.ac.in

**Abstract**

Floods are among India's most frequently occurring natural disasters, which disrupt all aspects of socio-economic well-being. A large population is affected by floods during almost every summer monsoon season in India, leaving its footprint through human mortality, migration, and damage to agriculture and infrastructure. Despite the massive imprints of floods, sub-basin level flood risk assessment is still in its infancy and requires advancements. Using hydrological and hydrodynamical models, we reconstructed sub-basin level observed floods for the 1901-2020 period. Our modelling framework includes the influence of 51 major reservoirs that affect flow variability and flood inundation. Sub-basins in the Ganga and Brahmaputra River basins witnessed substantial flood inundation extent during the worst flood in the observational record. Major floods in the sub-basins of the Ganga and Brahmaputra occur during the late summer monsoon season (August-September). Beas, Brahmani, upper Satluj, Upper Godavari, Middle and Lower Krishna, and Vashishti sub-basins are among the most influenced by the dams, while Beas, Brahmani, Ravi, and Lower Satluj are among the most impacted by floods and the presence of dams. Bhagirathi, Gandak, Kosi, lower Brahmaputra, and Ghaghara are India's sub-basins with the highest flood risk. Our findings have implications for flood risk assessment and mitigation in India.

## 1. Introduction

Flood risk to both natural and human systems is projected to increase due to climate change (IPCC, 2014, 2022). Extreme weather and climate extremes have increased under warming climate, leading to an increased frequency of natural hazards like floods, droughts, heat waves, cyclones, and heavy rains. Hydroclimatic extremes affect humans and infrastructure (Eidsvig et al., 2017; Peduzzi et al., 2009). Due to high vulnerability and lower adaptive capacity, developing countries are often the most impacted by extreme weather events. Further, developing countries usually take longer to recover from the hazards due to low climate resilience. Globally, floods are among the most devastating natural hazards (Ghosh & Kar, 2018). Among all flood types, riverine floods occur most frequently (Kimuli et al., 2021) and often cause substantial damage to agriculture and infrastructure. A considerable fraction of the population and infrastructure are exposed to flooding, which will also increase due to the projected increase in the magnitude and frequency of floods (Winsemius et al., 2018).

The increase in flood magnitude due to the warming climate has resulted in considerable economic losses (C. M. R. Mateo et al., 2014; Willner et al., 2018). The total financial loss will likely increase by 17% in the next 20 years due to climate change (Willner et al., 2018). Besides agriculture, floods significantly affect the built environment and transportation infrastructure (Kalantari et al., 2014). For instance, more than 7% of road and railway assets

globally are exposed to a 100-year return period flood (Koks et al., 2019). In Asia, about 75% of the population
is exposed to riverine floods (Varis et al., 2022). India falls among the top ten most flood-affected countries in
Asia and the Pacific (Kimuli et al., 2021). In addition, India is also among the top-ten countries that experienced
the highest human mortality due to floods. Considerable population exposure, climate change, and rapid growth
and development in flood-prone areas contribute to increased losses from floods.
In India, state administration takes decisions to mitigate floods while the central government provides financial
aid under severe conditions (Jain et al., 2017). The state authorities develop action plans to minimize flood
damage. Therefore, identifying the regions with higher flood risk is essential for planning and mitigation. Flood
impacts can be quantified according to the affected population, gross domestic product (GDP), and agricultural
practices (Ward et al., 2013). The flood risk assessment framework suggested by the Intergovernmental Panel on
Climate Change (IPCC) has been extensively applied at the regional and global scales (Allen et al., 2016; IPCC,
2014; Roy et al., 2021). The risk can be quantified as a function of vulnerability, hazard, and exposure (IPCC,
2014). To control the risk, reducing vulnerability is considered a short to the mid-term goal (V. Mishra et al.,
2022), while reducing hazards and exposure are long-term goals (Birkmann & Welle, 2015). Flood risk assessment
can assist in identifying the regions at high risk due to higher vulnerability, hazard, and exposure, which can be
used for developing a framework, methodology, and guidelines for flood mitigation and damage assessment.
A flood risk assessment performed on a global scale may not help in identifying the flood risk-prone regions at a
country scale due to the coarser spatial resolution (Bernhofen et al., 2022). Due to complex geomorphological
characteristics and diverse climatic conditions, India is considered a relatively high flood-risk region (Hochrainer-
Stigler et al., 2021). Therefore, estimating flood risk on a finer scale (e.g. sub-basin level) is essential for reliable
flood risk assessment. There have been studies on regional or river basin scales (Allen et al., 2016; Ghosh & Kar,
2018; Roy et al., 2021); however, those do not provide flood risk at a sub-basin scale in India. In addition, the
impact assessment of floods on transport infrastructure (rail and road infrastructure) still needs to be improved in
the country (Pathak et al., 2020; P. Singh et al., 2018). In addition, the role of dams and reservoirs in the flood
risk assessment should be addressed (Hirabayashi et al., 2013; Yamazaki et al., 2018a). Dams and reservoirs
considerably influence streamflow variability and can attenuate flood peaks (Dang et al., 2019; Vu et al., 2022;
Zajac et al., 2017). In contrast, dam operations and decisions can also worsen the flood situation in the downstream
regions. For instance, recent flooding in Kerala and Chennai was partly attributed to reservoir operations (V.
Mishra & Shah, 2018). India has more than 5300 large dams regulating river flow (National Register of Large
Dams (NRLD), 2019), affecting ecosystems, natural resources, and livelihoods (Acreman, 2000). Reservoirs
impact flow regulation, magnitude, timing, and extent of flooding in the downstream regions. Therefore, flood
risk assessment without considering the role of reservoirs can be inappropriate in the basins that are highly affected
by the presence of dams.
We use the H08 (Hanasaki et al., 2018) global hydrological model combined with the CaMa-Flood (Yamazaki et
al., 2011) model for the sub-basin level flood risk assessment in India considering the role of reservoirs. The
CaMa-Flood model combined with the H08 model has been used for several river basins globally (Boulange et
al., 2021; C. M. R. Mateo et al., 2013). The CaMa-Flood model performs well in simulating flood dynamics
(Chaudhari and Pokhrel, 2022; H. Dang et al., 2022; Gaur & Gaur, 2018; Hirabayashi et al., 2013, 2021; Yamazaki
et al., 2018; Yang et al., 2019). The CaMa-Flood model takes runoff as input simulated from any hydrological

model and can simulate flood depth and inundation. In India, almost all the major rivers are influenced by reservoirs (Lehner et al., 2011). Therefore, the major scientific questions that we address are: 1) How does the flood risk vary at the sub-basin level in India during the 1901-2020 period? 2) Which are the sub-basins where the presence of reservoirs considerably influences the flood risk? To address these questions, we use long-term observations (1901-2020) from India Meteorological Department (IMD) along with a hydrological modelling framework.

## 2. Data and Methods

### 2.1 Datasets

We used observed gridded precipitation (Pai et al., 2014) and daily maximum and minimum temperatures (Srivastava et al., 2009) from India Meteorological Department (IMD). We obtained gridded daily precipitation at 0.25° from IMD for the 1901-2020 period that was developed using station-based rainfall observations from more than 6900 gauge stations (Pai et al., 2014). The gridded rainfall product has been widely used for hydrological studies (Kushwaha et al., 2021; Shah & Mishra, 2016) and it captures the key features of the summer monsoon variability and orographic rainfall over the western Ghats and foothills of the Himalayas. We obtained daily 1° gridded maximum and minimum temperatures from IMD (Srivastava et al., 2009). The gridded temperature dataset is developed using observations from 395 stations located across India. Bilinear interpolation was used to convert the 1° gridded temperature to 0.25° resolution to make it consistent with the gridded precipitation. For the regions outside India, we obtained observational meteorological datasets (rainfall and temperature) at 0.25 degrees from Princeton University (Sheffield et al., 2006). Gridded datasets from Sheffield et al. (2006) compare well against the IMD observations and have been used in hydrological applications in India (Shah & Mishra, 2016).

Observed daily streamflow at gauge stations and reservoir live storage were obtained from India Water Resources Information System (India-WRIS). We considered the influence of 51 major reservoirs located in different river basins to examine the impact of reservoirs on floods using the CaMa-Flood model (Figure S1). The information of dams was obtained from the National Register of Large Dams (NRLD) [Table S1]. We used the Global Surface Water (GSW) extent to estimate flood occurrences at a monthly timescale (Pekel et al., 2016). Simulated flood occurrences during the period of the GSW database (1985-2020) were used to validate the performance of the hydrological model in simulating flood extent (Pekel et al., 2016). In addition, we obtained reported flood details from the Emergency Events Database (EM-DAT, http://www.emdat.be/) and Dartmouth Flood Observatory (DFO, http://floodobservatory.colorado.edu/). EM-DAT is developed by the Centre for Research on the Epidemiology of Disasters (CRED), while the University of Colorado manages DFO. We used population data from Global Human Settlement Layers (GHLS) to estimate flood exposure. Finally, we used roadway and railway network data to assess the impact of floods on the infrastructure.

### 2.2 H08-CaMa-Flood combined model

We used the H08 (Hanasaki et al., 2018) global hydrological model to simulate hydrological variables. The H08 is a distributed global water resources model comprising six sub-models: land surface hydrology, river routing, reservoir operation, crop growth, environmental flow, and water abstraction. The model estimates baseflow using a leaky bucket method, while runoff is calculated based on saturation excess non-linear flow (Hanasaki et al.,

2008). The H08 model can be run separately or combined with any hydrodynamic model to perform flow routing.
The H08 model uses precipitation, air temperature, short and longwave radiations, wind speed, surface pressure,
and specific humidity as input meteorological forcing. Soil parameters for the H08 model were obtained from
Harmonized World Soil Database (HWSD). We forced the H08 model with the input meteorological forcing at
0.25° spatial and daily temporal resolution. We combined the H08 land surface model with the CaMa-Flood
model. The CaMa-Flood model has been previously combined with the H08 model to obtain flood inundation
estimates (C. M. Mateo et al., 2014).
The CaMa-Flood (version 4.1) is a hydrodynamic model (Yamazaki et al., 2011), which simulates river-floodplain
dynamics (Yamazaki et al., 2013). The CaMa-Flood model has been extensively used for better performance in
simulating discharge and flood peaks (Zhao et al., 2017). The CaMa-Flood model considers the role of dams and
reservoirs for streamflow and flood inundation simulations (Chaudhari & Pokhrel, 2022; C. M. Mateo et al., 2014;
Pokhrel et al., 2018). We ran the CaMa-Flood model at a finer spatial resolution (0.1°) using the H08-simulated
runoff (0.25°) as input. We calibrated the combined model (H08 and CaMa-Flood) for India's eighteen major river
basins for at least one gauge station each, considering the influence of 51 major dams. The gauge stations were
selected in the farthest downstream of the river basin based on the availability of observed streamflow. The
influence of reservoir operations was simulated using the CaMa-Flood model and evaluated against the observed
daily live reservoir storage.
Large-scale global hydrological models do not perfectly capture the observed trends and variations as these are
often not well calibrated at river basin scale (Krysanova et al., 2018). The H08 model performs well when
calibrated at the river basin scale rather than coarser domains such as climate zones (Chuphal & Mishra, 2023;
Yoshida et al., 2022). Here, we manually calibrated the H08 model by adjusting four key parameters that
considerably influence streamflow for each river basin, which include single-layer soil depth, gamma, bulk
transfer coefficient, and tau (Hanasaki et al., 2008; Raghav & Eldho, 2023). A more detailed discussion about the
calibration parameters of H08 are discussed in Dangar & Mishra (2021). Different sets of combinations of
calibration parameters within a range were used to calibrate the H08 model. The employed sets of parameters for
the 18 river basins in the Indian sub-continent are listed in Table S2. The calibrated parameters account for the
effect of human interventions because the model calibration is performed against the observed streamflow rather
than the naturalized streamflow (Duc Dang et al., 2020). We evaluated the model performance using the
coefficient of determination ($R^2$) and Nash-Sutcliffe Efficiency (NSE) for daily streamflow and reservoir live
storage. In addition, we compared the simulated and satellite-based observed flood occurrences. The satellite-
based flood occurrence is calculated using the Global Surface Water (GSW) dataset (Pekel et al., 2016), available
for the 1984-2020 period. We forced the well-calibrated combined (H08 and CaMa-Flood) models with observed
meteorological forcing from India Meteorological Department (IMD) at 0.25° spatial resolution to conduct
simulations from 1901 to 2020. The H08 model simulated runoff is used in CaMa-Flood to rout flood dynamics
at six arc-minutes (0.1 degrees). We generated the flood depth maps for the historical worst flood at the sub-basin
level. The worst flood is based on the highest magnitude of river flow observed at the subbasin outlet. The
generated flood depths at 6 arc-minutes (0.1°) were further downscaled to 1 arc-minute (~0.185 km) resolution
using the downscaling module available within the CaMa-Flood.
We used C-ratio (Nilsson et al., 2005; Zajac et al., 2017) to assess the potential impact of dams along a river. The
C-ratio is an identifier calculated as the ratio of total maximum storage capacity of the upstream reservoirs to the
mean annual discharge at a gauge station in the downstream region (Nilsson et al., 2005; Zajac et al., 2017). We
calculated the C-ratio at the outlets of each sub-basins that are influenced by the presence of dams. A C-ratio of
less than 0.5 indicates that the sub-basin is minimally affected by the presence of dams. Further, to identify sub-
basins susceptible to flood inundation resulting from dam operations, we multiplied the percentage of flooded
area in each sub-basin by its corresponding C-ratio. This enabled us to identify the sub-basins that experience
substantial flood inundation and are considerably impacted by the presence of reservoirs. Finally, we estimated
the exposed rail and road infrastructure affected by floods. The flooded area overlapped over the road and railway
network to estimate the network length affected by floods in a sub-basin. We considered the flooded area of the
observed worst flood. The subbasins with the highest rail and road infrastructure exposure to floods were
identified.
**2.3 Risk assessment**
We estimated flood risk using hazard, exposure, and vulnerability based on the common framework adopted by
the United Nations in the Global Assessment Reports of the United Nations Office for Disaster Risk Reduction
(UNISDR, 2011, 2013). A similar framework was used in previous studies for flood risk assessments (C. M. R.
Mateo et al., 2014; Tanoue, 2020; Winsemius et al., 2013). We multiplied the normalized values of hazard,
exposure, and vulnerability to estimate the risk as:

$Risk = Vulnerability * Exposure * Hazard$                … … (1)

The flood risk assessment can help identify the hotspots and prioritize climate adaptation (de Moel et al., 2015).
Among the three components, vulnerability is a degree of damage to a particular object at flood risk with a
specified amount and present on a scale from 0 to 1. We obtained the vulnerability index for each district from
the "Climate Vulnerability Assessment for Adaptation Planning in India Using a Common Framework", a report
developed        by        the        Department        of        Science        and        Technology
(https://dst.gov.in/sites/default/files/Full%20Report%20%281%29.pdf). The vulnerability of each district is
calculated using 14 indicators, each with equal weights. The indicators capture both sensitivity and adaptive
capacity. We estimated the vulnerability index of each sub-basin by taking the spatial mean of the vulnerability
of the districts falling into the sub-basins. Exposure is termed as assets and population in a flood-exposed area
resulting in flood damage (Marchand et al., 2022). The population dataset is a critical component in performing
exposure estimation. The exposure is defined as the fraction of the population exposed to the flood extent (Smith
et al., 2019). We completed the flood exposure estimate using the Global Human Settlement Layers (GHSL)
population dataset (Joint Research Centre (JRC) et al., 2021), which is available at a resolution of 30 arc-seconds
for 1975, 1990, 2000, 2014 and 2015. We used the population data for the year 2015 throughout this study. We
rescaled the population data to 6 arc-minutes to make it consistent with the flooded area simulated from the
combined model. We estimated the hazard as the exceedance probability of a flooded area exceeding half of the
historical maximum flooded area in the last 50 years. We used normalized vulnerability, exposure, and hazard to
estimate the risk.
**3.  Results**

### 3.1 Calibration and evaluation of hydrological models

We calibrated and evaluated the performance of the H08 and CaMa-Flood combined models against the observed daily streamflow (Figure 1). Due to the unavailability of daily observed streamflow for the three transboundary river basins (Indus, Ganga and Brahmaputra), we used observed monthly streamflow to calibrate the model. In addition, we evaluated the model performance for daily live storage of the 51 reservoirs after the calibration against the observed flow (Figure 1). The model exhibited good skills ($R^2 > 0.6$ and NSE > 0.6) for almost all the river basins except Cauvery, East Coast, Northeast Coast, and Sabarmati. The model also performed well with NSE greater than 0.6 for more than 80% of the selected reservoirs in simulating daily live storage for the selected reservoirs. We estimated the bias and timing error in simulating peak discharge at all the selected gauge stations (Figure S2). We calculated the bias in the model simulated annual maximum streamflow against the observed annual maximum streamflow for the time periods for which observations are available. We excluded the transboundary rivers (Ganga, Brahmaputra and Indus) as timing error (in days) could not be estimated due to the unavailability of daily observed flow. While other gauge stations exhibited moderate bias, gauge stations in Cauvery, Sabarmati, and Mahi rivers basins show a considerable dry bias. Contrary to several other stations where the mean timing error was below two days, the Sabarmati river basin displayed a comparatively higher mean timing error. The relatively poor performance of the model in these river basins can be attributed to the lack of long-term observations as well as substantial human interventions that can affect the observed flow.

We compared model-simulated, and satellite-based observed flood occurrence for the 1984-2020 period (Figure 2). In addition, we compared the model-simulated flood events against Sentinel-1 SAR and MODIS satellite-based imagery for a few flood events based on the satellite data availability (Figures. 3, S3, S4). We found that the model simulated flood extent captures the satellite based flood extent. However, we note that the model overestimated the flood extent in Ganga river basin and underestimated in Brahmaputra river basin, therefore, showing a non-systematic bias. Moreover, a considerable difference in the flood extent based on the two satellite datasets was observed, which highlights the observational uncertainty in the estimation of flood extent. In general, the model exhibits satisfactory performance in simulating flood extent against the satellite-based observations. However, the model overestimates flood extent in the Ganga basin, which could be attributed to the influence of cloud contamination and dense vegetation cover on satellite-based flood estimates (Chaudhari & Pokhrel, 2022). On the other hand, the model underestimates the flood occurrence in the upstream region of the Brahmaputra River. This could be due to limitations in model parameterization, as observed flow is limited in the transboundary river basins. Despite the good performance against the observed streamflow, the simulated flood extent has a considerable bias, which can be attributed to satellite-based flood extent mapping limitations and the model's ability to capture the flood extent accurately. The model-simulated flood extent shows a good agreement against the reported flood from EM-DAT and DFO databases (Figure S5). In addition, the simulated flood extent also showed a good agreement with the reported flood in cities in the Brahmaputra and Ganga River basins. Given the limitation in the streamflow and flood extent observations, the hydrological models perform satisfactorily and can be used for the sub-basin level risk assessment.

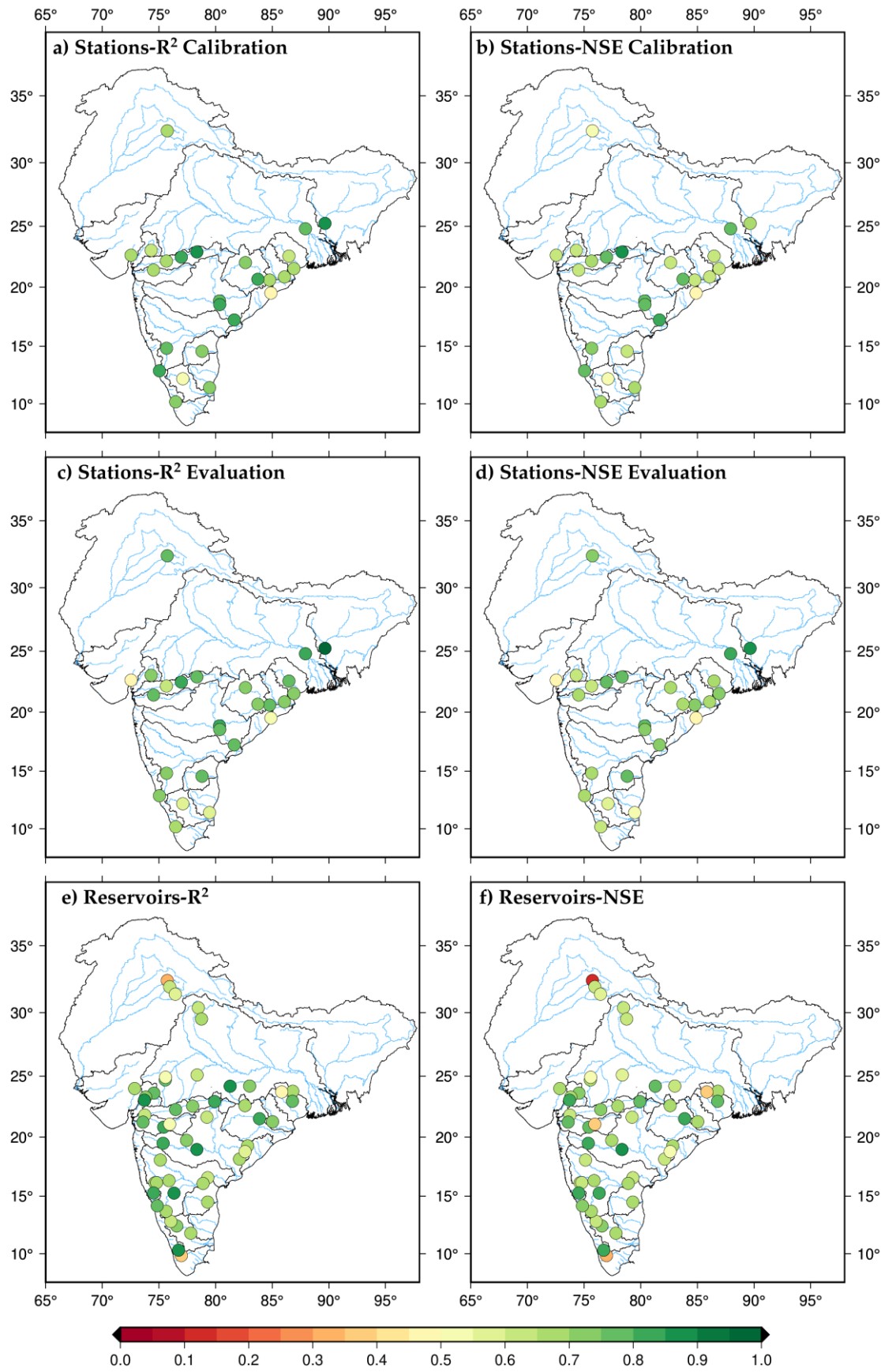


**Figure 1: Calibration and evaluation of the combined model for daily river flow and reservoir storage at gauge stations and daily live storage of reservoirs**



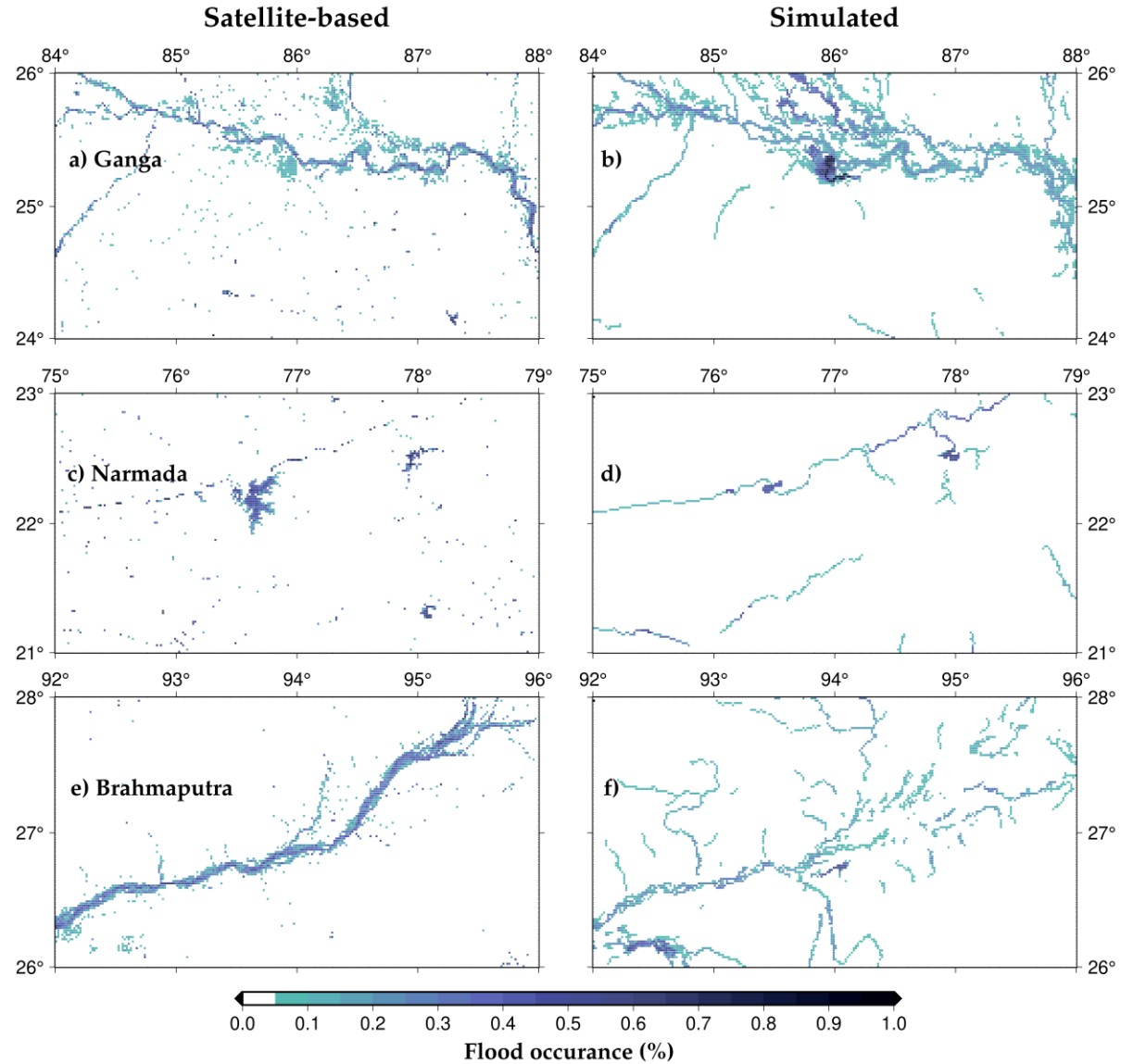


**Figure 2: Simulated flood occurrences compared with satellite-based flood occurrence for different regions in Ganga, Narmada and Brahmaputra River basin.**

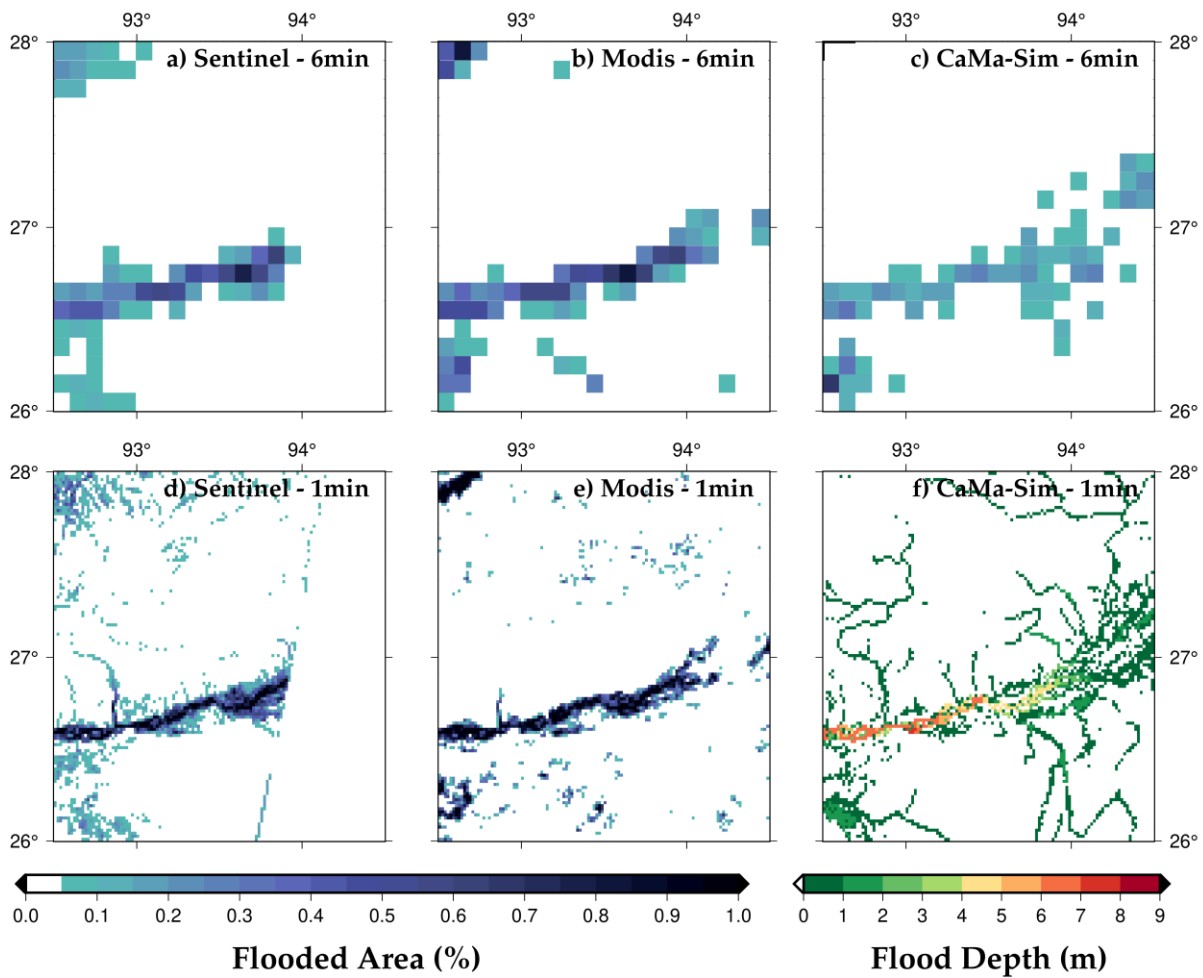

**Figure 3: Simulated flood extent compared with Sentinel-1 SAR and MODIS satellite-based flood extent for the 2016 flood event in the Brahmaputra river**

**3.2 Estimation of the observed flood extent**

Next, we reconstructed the flood inundation for the observed worst flood for each sub-basin for the 1901-2020 period in India. The inundation extent for the worst flood can help us identify the sub-basin with higher flood risk. We estimated flood depth and inundated area for each sub-basin for the worst flood during the last 120 years (Figure 4). In addition, we identified the occurrence of the worst flood at the sub-basin level during the 1901-2020 period. We highlighted ten sub-basins that experienced the highest fractional area affected by the worst flood. Sub-basins in the Ganga and Brahmaputra rivers are among the most highly influenced by the worst flood. For instance, Ghaghra, Kosi, Bhagirathi, Gandak, Gomti, lower Sabarmati, upper Yamuna, Ramganga, and Baitarani sub-basins had the highest fractional area affected by the worst flood during 1901-2020 (Figure 4). The fractional area of sub-basins in the semi-arid western India is less affected compared to those located in the Ganga basin. For example, the lower Sabarmati sub-basin of the Sabarmati River basin is among the sub-basins that are highly influenced by the observed worst flood. We also find that the worst flood in the same year did not affect all the sub-basins within a river basin (Figure S6). For instance, all the highly influenced sub-basins experienced the worst flood in different years in the Ganga basin (Figure 4). Most of the top flood-affected sub-basins experienced floods during August-September in the summer monsoon season. Overall, the flood extent due to the worst flood

is substantially greater in the sub-basins of the Ganga and Brahmaputra river basins compared to other basins in
India (Figure 4). Ganga river basin also has the highest population density among all the basins in the Indian sub-
continent, which makes it vulnerable for the flood risk.

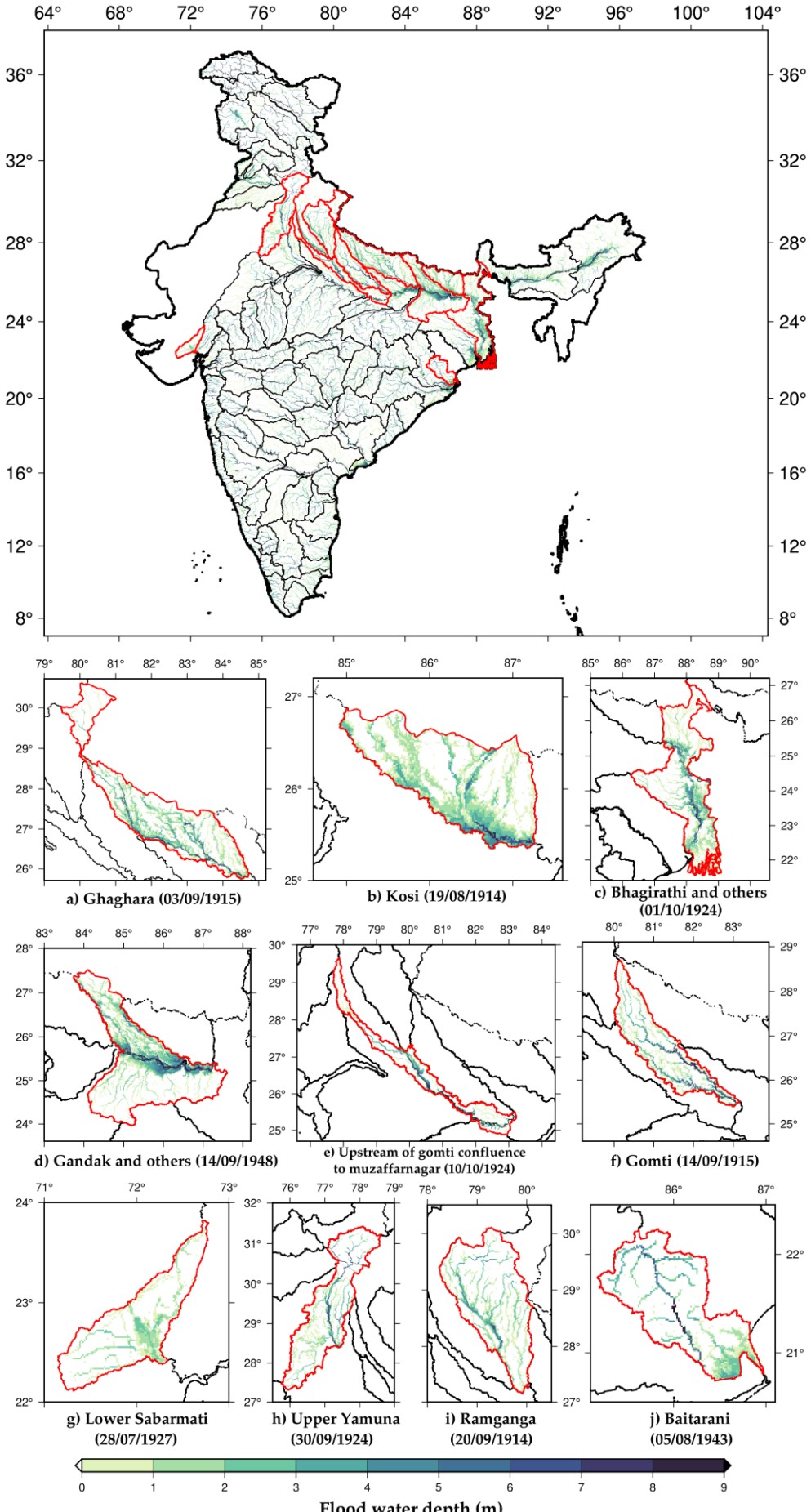

a) Ghaghara (03/09/1915)

b) Kosi (19/08/1914)

c) Bhagirathi and others
(01/10/1924)

d) Gandak and others (14/09/1948)

e) Upstream of gomti confluence
to muzaffarnagar (10/10/1924)

f) Gomti (14/09/1915)

g) Lower Sabarmati
(28/07/1927)

h) Upper Yamuna
(30/09/1924)

i) Ramganga
(20/09/1914)

j) Baitarani
(05/08/1943)

Flood water depth (m)


**Figure 4: Flood depth map for the observed worst flood for each sub-basins, highlighting the sub-basins with maximum flood inundated area (%) (a) Ghaghara – Ganga River basin (b) Kosi – Ganga River basin (c) Bhagirathi and others – Ganga River basin (d) Gandak and others – Ganga River basin (e) Upstream of Gomti confluence to Muzaffarnagar – Ganga River basin (f) Gomti – Ganga River basin (g) Lower Sabarmati – Sabarmati River basin (h) Upper Yamuna – Ganga River basin (i) Ramganga – Ganga River basin (j) Baitarani – Brahmani River basin**

Next, we examined the precipitation, streamflow, and flood-affected area (%) for the ten sub-basins that had the highest fractional flood affected area for the worst flood during 1901-2020 (Figure 5). As floods mostly occur during the summer monsoon season in India (V. Mishra et al., 2022; Nanditha & Mishra, 2021), we examined the temporal variability of precipitation, and streamflow during the monsoon season of the worst flood year. Nanditha and Mishra (2022) reported that multi-day precipitation is India's most robust driver of floods. Moreover, extreme precipitation and wet-antecedent conditions trigger floods in India (Nanditha & Mishra, 2022). We find that the Ghaghara sub-basin of the Ganga river experienced the worst flood in September 1915, affecting more than 10,000 km$^2$ area of the sub-basin. A multi-day rainfall in late August and early September (1915) caused the worst flood in the basin. The Kosi sub-basin of the Ganga river experienced the worst flood in August 1914, which affected more than 5000 km$^2$ of the basin (Figure 5). Similarly, Bhagirathi and other sub-basins in the Ganga river basin were affected by the worst flood in late September 1924, which inundated more than 12000 km$^2$ of the sub-basin. Similarly, Gandak and Gomti river basins experienced the worst floods in 1948 and 1915, respectively. Our results agree with the information presented in previous studies (Agarwal & Narain, 1991; Fredrick, 2017; Joshi, 2014; D. K. Mishra, 2015; A. Singh et al., 2021). We find that most of the sub-basins of the Ganga river basin are prone to large extents of flood inundation. Moreover, the worst floods in most sub-basins were caused by multi-day precipitation, a prominent driver of floods in the Indian sub-continental river basins (Figure 5).

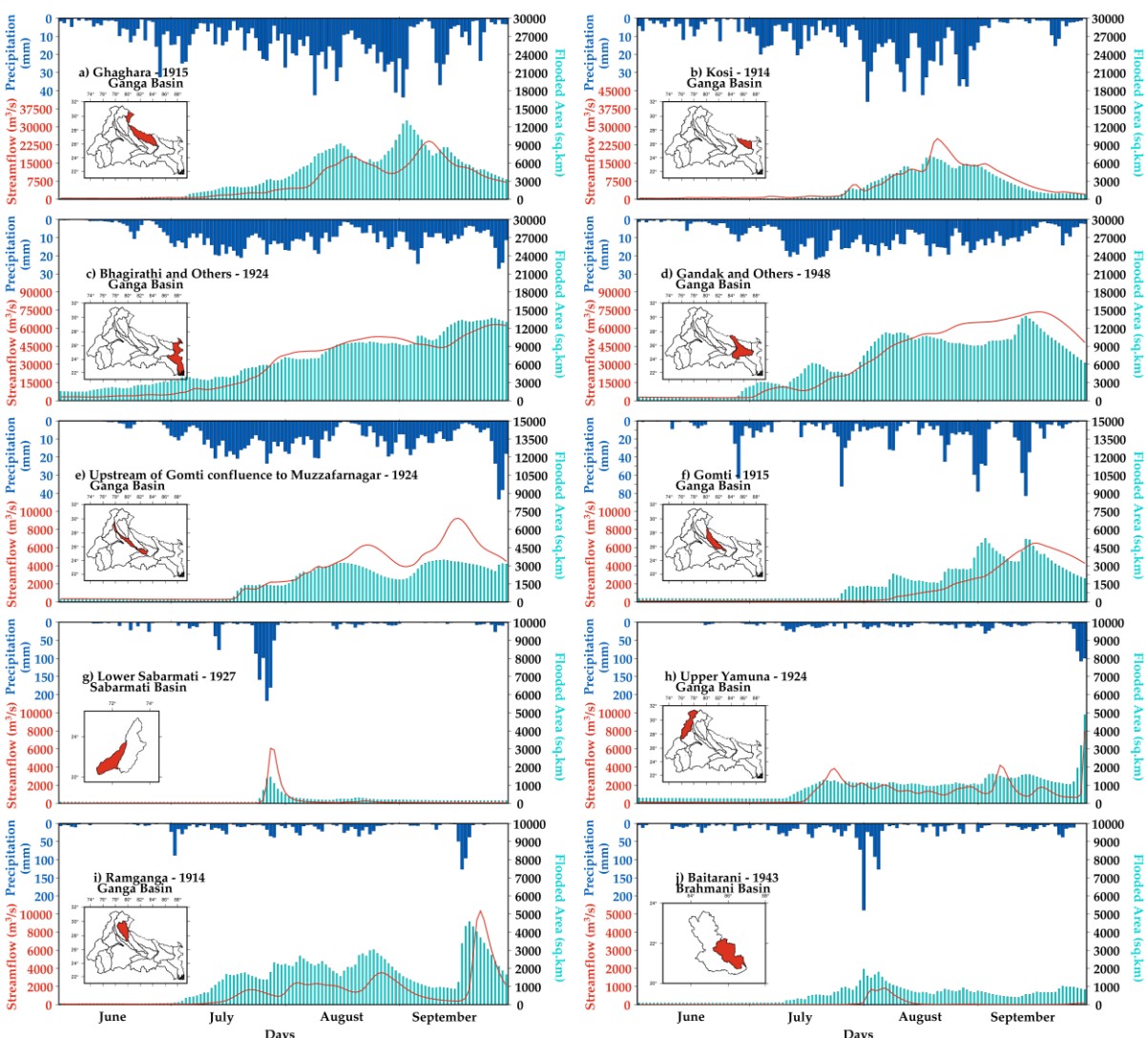

**Figure 5: Daily upstream precipitation (mm, blue), the H08 model simulated streamflow (red) at the sub-basin outlet (m3/s), and flooded area (km2, green) for the summer monsoon (June-September) period of the corresponding worst flood year. (a) Ghaghara - Ganga River basin (b) Kosi - Ganga River basin (c) Bhagirathi and others - Ganga River basin (d) Gandak and others - Ganga River basin (e) Upstream of Gomti confluence to Muzaffarnagar - Ganga River basin (f) Gomti - Ganga River basin (g) Lower Sabarmati – Sabarmati River basin (h) Upper Yamuna – Ganga River basin (i) Ramganga – Ganga River basin (j) Baitarani – Brahmani River basin**

To further examine the flood-affected area at the sub-basin level, we estimated the mean annual maximum flooded area (Figure 6a) and historical maximum flooded area using the H08-CaMa flood models (Figure 6b). Most of the highly flooded sub-basins are in the Ganga River basin. While the mean annual maximum flooded area for the top flood-affected sub-basins ranged between 10 to 15%, their maximum flooded area varied between 30 to 40%. Other than sub-basins from the Ganga river basin, Baitarani, lower Tapi, lower Godavari, Brahmani, and lower Mahanadi also showed a considerable mean flooded area during the 1901-1920 period. In the case of the maximum flooded area, Gandak, Kosi, and Ghaghara confluence to Gomti confluence sub-basins exhibited more than 20% flooded area. Sub-basins from the other river basins, such as lower Tapi, lower Narmada, Baitarani, and lower

Satluj, are in the top fifteen sub-basins with the highest flooded area. The sub-basins in the Ganga and
Brahmaputra rivers are the most flood-affected. Moreover, the Ganga and Brahmaputra rivers experience the
highest floods among all the river basins (Mohanty et al., 2020; Mohapatra & Singh, 2003).

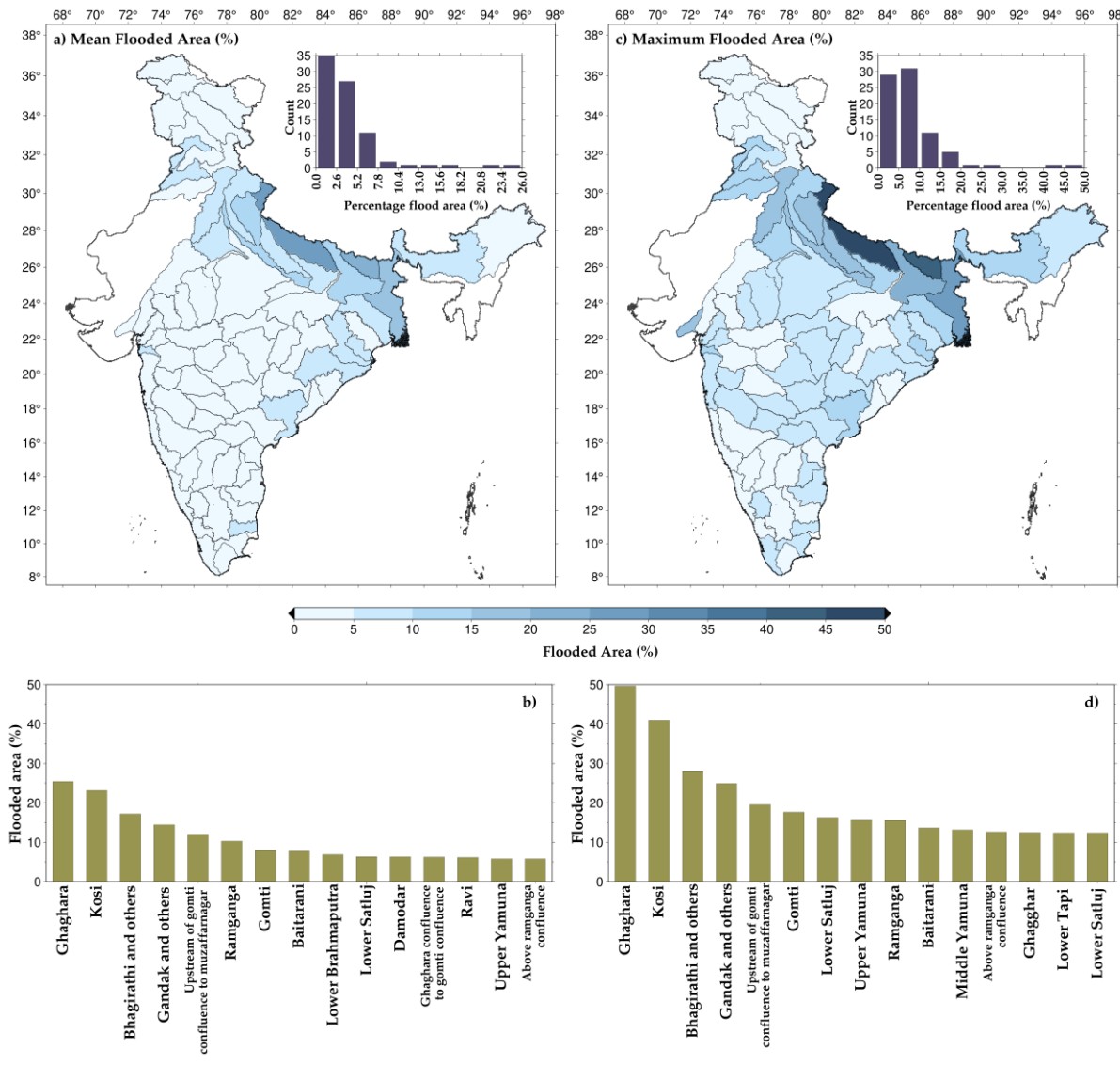


**Figure 6: (a) Mean of annual maximum flooded area (percentage) between 1901-2020 and the overall**
**distribution (b) highlighting the top fifteen sub-basin. (c) Historical maximum flooded area (percentage)**
**and the overall distribution (d) highlighting the top fifteen sub-basin.**
**3.3 Influence of reservoirs on flood extent**
We selected and considered 51 major reservoirs to examine their influence on flood risk based on the availability
of the observed storage data. We estimated C-ratio for each sub-basin considering the river flow at the outlet to
investigate the impact of reservoirs on streamflow. C-ratio can vary between zero to infinity, and higher values
indicate the prominent effect of dams on river flow. We identified sub-basins with a greater influence on dams
based on the C-ratio. We find that Beas, Brahmani, upper Satluj, Upper Godavari, Middle and Lower Krishna,
and Vashishti are among the most influenced by the dams. Beas sub-basin has the highest C-ratio (4.16) among
all the sub-basin in the Indian sub-continent (Figure 7a). Out of the 80 sub-basins, only eleven have C-ratio greater

than 0.5. 64 out of 80 sub-basins have a C-ratio between zero to 0.42 (Figure 7a). We considered only 51 major reservoirs in our analysis. However, there are several major and minor dams for which observed data is unavailable. Therefore, the influence of reservoirs based on the C-ratio might need to be considered. However, our analysis indicates that dams in a few sub-basins can significantly alter the river flow and flood risk. For instance, dams effectively alter extreme flow's timing, duration, and frequency (Mittal et al., 2016). C-ratio alone may not effectively capture the influence of dams on floods; therefore, we multiplied the fractional area affected by floods and the C-ratio for each sub-basins. For instance, if a sub-basin is considerably affected by dams and has a large flood extent, the value of the multiplied ratio will be higher. The multiplier ratio can effectively identify the sub-basins with high flood-affected areas and flow regulated by the reservoirs. We find that Beas, Brahmani, Ravi, and Lower Satluj are among the highly influenced by floods and the presence of reservoirs. Overall, the sub-basins with higher C ratio and the highest flood-affected area are across the Indian subcontinent. Central India has sub-basins that are relatively less affected by floods and the presence of dams.

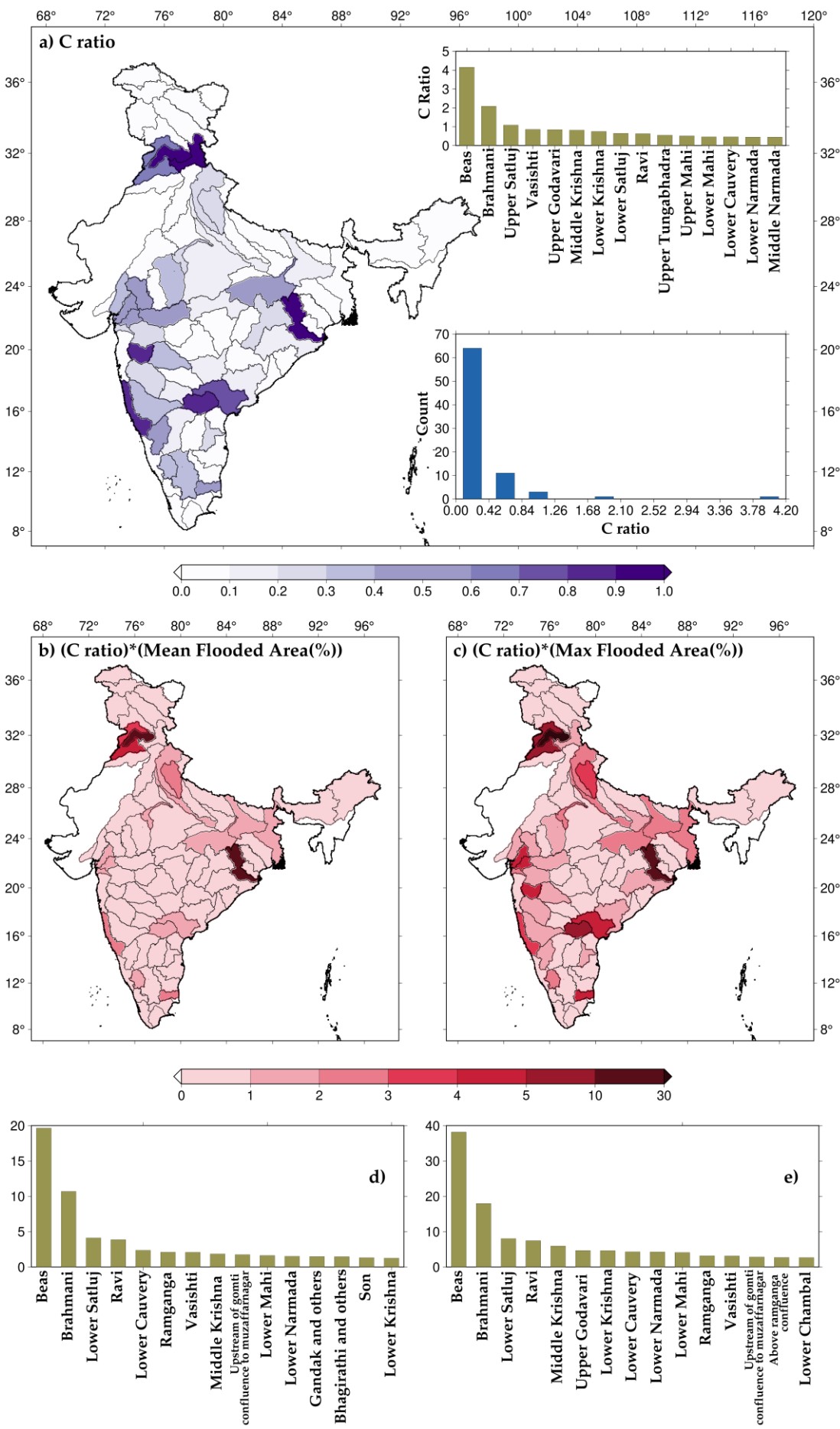

**Figure 7: (a) Sub-basin wise C-ratio, top fifteen sub-basins and distribution of sub-basins based on C-ratio values (b) Mean of annual maximum flooded area (percentage) multiplied with C-ratio (d) highlighting top 15 sub-basins (c) Historical maximum flooded area (percentage) multiplied with C-ratio (e) highlighting top 15 sub-basins.**

**3.4 Sub-basin level flood risk assessment**

Next, we identified the roads (national highways) and railway exposure to riverine floods for each subbasin. Climate change will adversely affect rail and road networks (Hooper & Chapman, 2012; Padhra, 2022). A considerable length of roads is affected due to surface flooding resulting from high-intensity rain (Koks et al., 2019). Therefore, we examined the impact of floods on rail and road infrastructure in India. We estimated the length of the road and railway network potentially affected by the worst flood that occurred during 1901-2020. We overlapped the road and rail network over the flooded area and estimated the network length exposed to floods (Figures 8a-b). The estimated length for each sub-basin was normalized between zero and one (Figures 8c-d). We find that the road network can be the most affected by the floods in the Gandak, Kosi and Ghaghara confluence to Gomti confluence in the Ganga river basin. On the other hand, a considerable part of the rail network can be affected by floods in Son, Kosi, and Upper Yamuna subbasins. Moreover, in Bhagirathi and Gandak river basins, more than 50 km of road network falls in the flood-prone regions (Figure 8e). There are ten sub-basins in which more than 20 km of road network falls in flood-prone areas of India. Similarly, over 20 km of the rail network is in the flood-affected areas of the six sub-basins (Upper Yamuna, Son, Kosi, Brahmani) [Figure 8f].

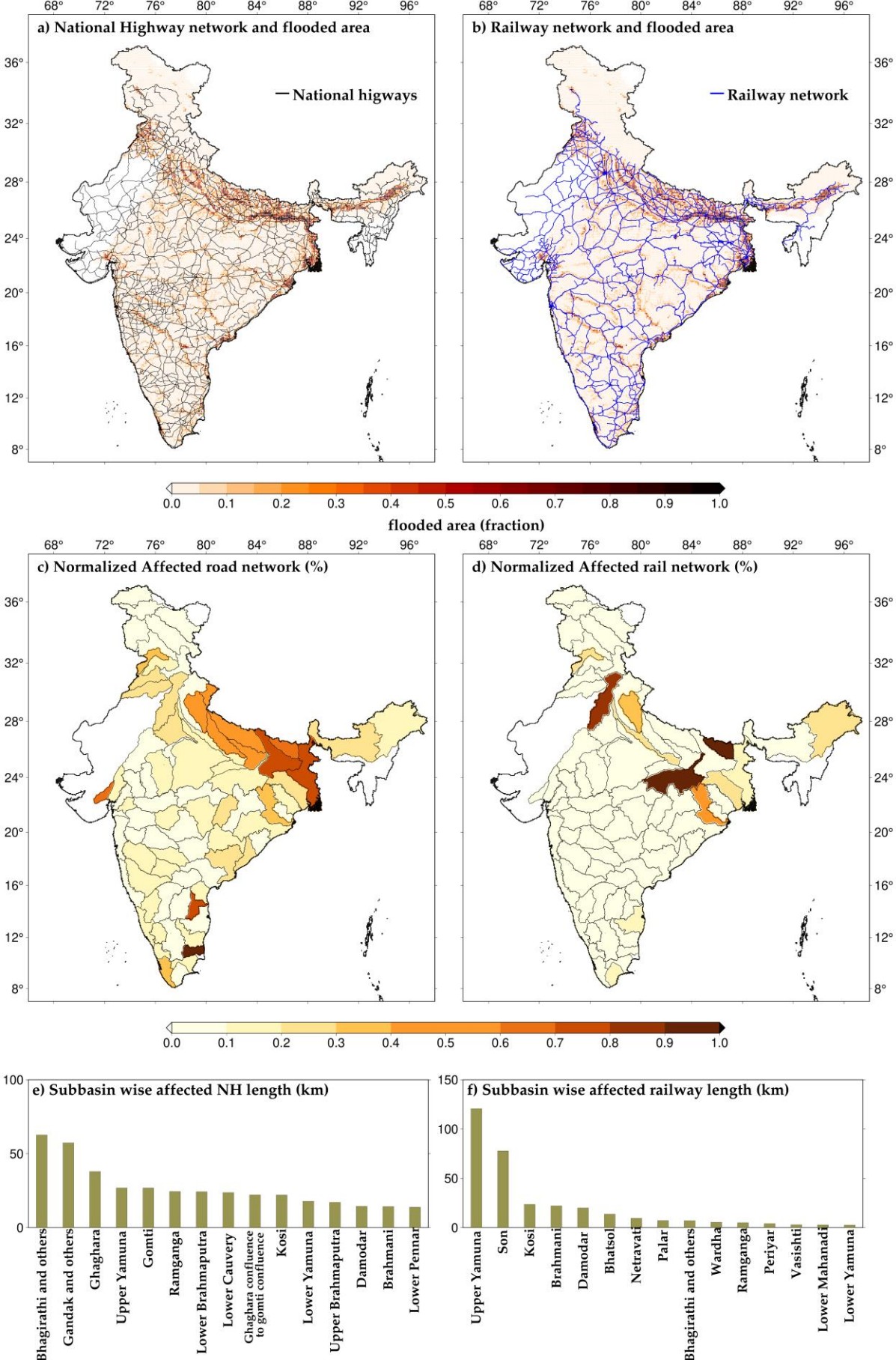

**Figure 8: Flood impacts on roads and railways infrastructure. (a-b) National Highways network and Railway network overlapped over the flooded area in worst flood cases, (c-d) subbasin wise normalised flood affected road and railway network (percentage), (e-f) top 15 subbasins with most affected national highways and railway length (km).**

Finally, we estimated sub-basin level flood risk using normalized vulnerability, hazard, and exposure (Figure 9). Vulnerability for each sub-basin in India was assessed using the national vulnerability assessment data available at the district level. We estimated hazard probability considering 50% of the inundated area for the worst flood as a benchmark. The likelihood of flood inundated areas in a sub-basin exceeding the benchmark was used in the risk assessment. Similarly, we used the worst flood extent and gridded population data to estimate flood exposure. The sub-basins in north-central India have a relatively higher vulnerability calculated using the socio-economic indicators. The vulnerability is relatively lower in north India and the Western Ghats. Kosi, Gandak, and Damodar sub-basins have the highest vulnerability. We find that hazard probability is higher in the sub-basins of Brahmaputra, rivers in the western Ghats, and a few sub-basins of the Indus river basin (Figure 9b). For instance, upper Satluj, Chenab, and Jhelum sub-basins of the Indus river have higher hazard probability. Other than the Western Ghats, most sub-basins in Peninsular India have relatively lesser hazard probability. Exposure, which represents the fraction of the population affected by flood under the worst flood scenario, is higher in the Indo-Gangetic Plain. Apart from the sub-basins of the Ganga River basin, the lower Brahmaputra, lower Godavari, and Baitarani sub-basin show higher exposure. Therefore, Ganga and Brahmaputra Rivers basins are the highest flood-prone river basins and have high flood exposure. Rentschler et al. (2022) also reported that the highest population exposure due to floods is in Uttar Pradesh, Bihar, and West Bengal, which is part of the Ganga river basin.

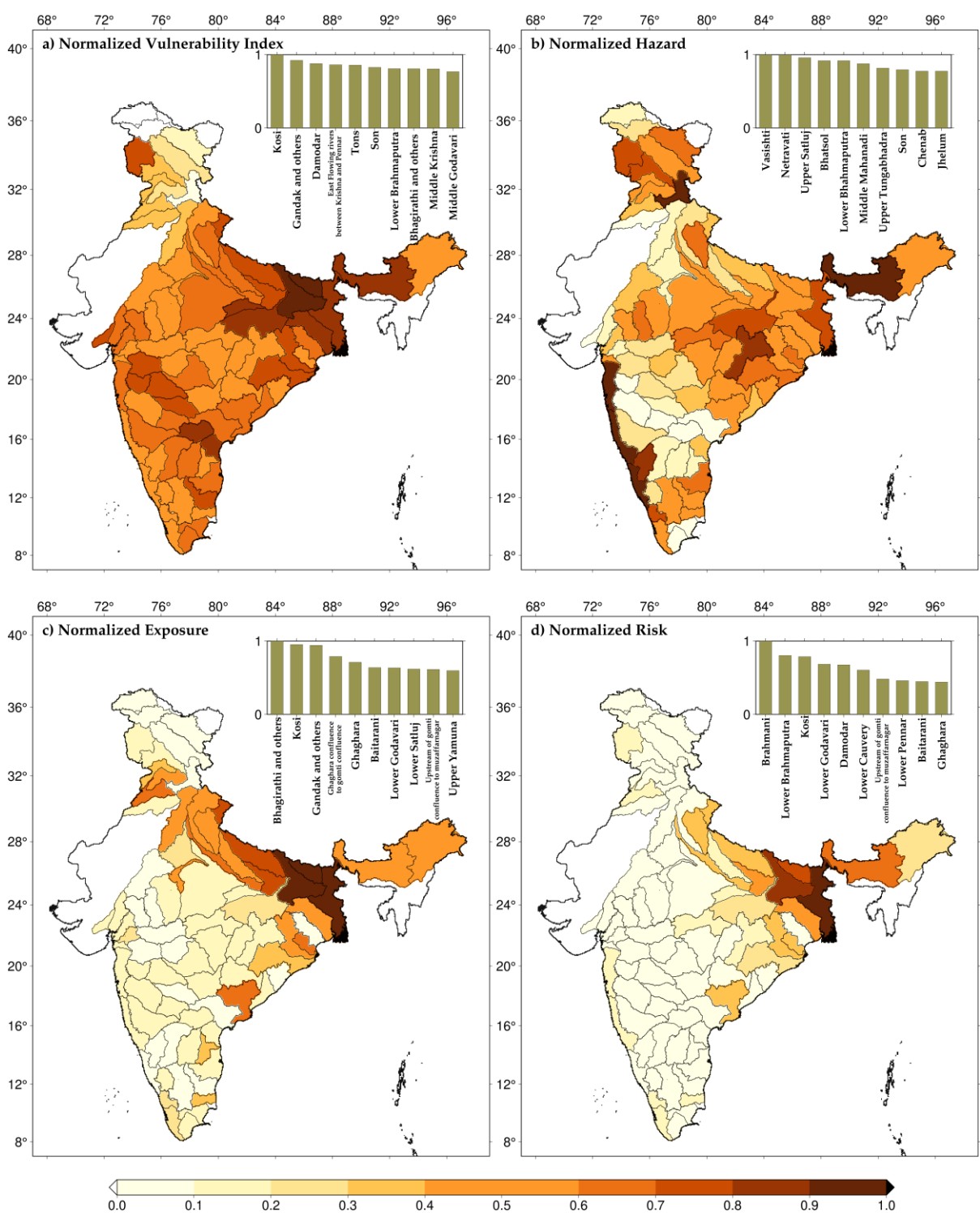

**Figure 9: Sub-basin level (a) Normalized vulnerability index (b) Normalized hazard (c) Normalized exposure (d) Normalized risk. The top 10 sub-basins are highlighted as bars in panels inside the figures.**

We estimated the flood risk for each sub-basin, a collective representation of vulnerability, hazard, and exposure. As expected, the flood risk is higher in the Ganga and Brahmaputra river basins compared to other parts of the country. The higher flood risk in these basins can be attributed to higher vulnerability, hazard probability, and exposure. For instance, Bhagirathi, Gandak, Kosi, lower Brahmaputra, and Ghaghra are the sub-basins with the highest flood risk in India (Figure 9d). Despite the higher hazard probability in the sub-basins of the Indus and

west coast river basins, the overall flood-risk is considerably lower than the sub-basins of the Ganga and Brahmaputra river basins primarily due to less vulnerability and exposure. Our results show that flood risk in some of the sub-basins of the Ganga and Brahmaputra river basins can be reduced by reducing the vulnerability.

**4. Discussion and conclusions**

Flood risk mapping is essential for risk reduction and developing mitigation measures. The flood risk will likely increase due to increased hazard probability and exposure (Ali et al., 2019). Hirabayashi et al. (2013) showed that a warmer climate would increase the risk of floods on a global scale. In India also, floods are expected to become more likely under warming climate. For instance, Ali et al. (2019) reported that multi-day floods are projected to rise faster than single-day flood events. The projected rise in the flood frequency in India can be attributed to increased extreme precipitation under warming climate (Mukherjee et al., 2018). Observational studies have also concluded that there has been a considerable rise in extreme precipitation in India during the summer monsoon season (Roxy et al., 2017), which is linked to warming climate. While the warming climate is directly linked to the increased frequency of extreme precipitation, its association with riverine floods is not straightforward. For instance, Nanditha & Mishra (2021, 2022) reported that multi-day precipitation on the wet antecedent condition is the most favourable conditions for riverine floods in India.

While mapping the flood risk at appropriate spatial resolution is complex and challenging, it is vital for disaster risk reduction. Flood inundation mapping that provides the spatial extent of flooding is crucial as the first responders use it during a flood emergency (Apel et al., 2009). There are several approaches to mapping flood inundation (Teng et al., 2017). Various hydrological models have been employed for conducting flood risk assessments at a global scale (Dottori et al., 2018; Gu et al., 2020; Tabari et al., 2021). For instance, Dottori et al. (2018) used the H08 model combined with CaMa-Flood model to estimate losses resulting from river flooding at the country level. Additionally, the LISFLOOD model (van der Knijff et al., 2010) at 5 km spatial resolution was used to estimate the river flood risk in Europe (Alfieri et al., 2018). Flood risk assessment at relatively larger scales are conducted using the coarse resolution land surface hydrological models. The objective of these large scale flood risk assessment is to identify regions that are flood-prone (Dottori et al., 2018; Gu et al., 2020; Tabari et al., 2021). On the other hand, high resolution flood inundation mapping is needed to understand the local flood risk and damage caused to infrastructure. For the analysis of flood inundation during a particular floodat a local scale, high-resolution models such as HEC-RAS and Mike FLOOD can be employed (Khalaj et al., 2021; Nguyen et al., 2016). High resolution flood risk mapping requires comprehensive information of high-resolution topography, cross-sections of channels, and data associated to structural measures of flood protection. However, the smallest subbasin considered in our study has more than 5000 km² area (Fig S7), while most subbasins have area between 10,000 and 50,000 km², with Lower Yamuna being the largest subbasin, with an area of 124,867.19 km². Therefore, the performance of our modelling framework against the satellite and other observations can be considered satisfactory to provide a sub-basin scale flood risk assessment. Moreover, we used hydrodynamic modelling to develop long-term flood inundation maps for the Indian sub-basins. The long-term data (1901-2020) provides us a record of several floods, which can help in robust estimates of flood risk in different sub-basins.

While high-resolution models are suitable for event-specific finer-level flood assessments, their feasibility
diminishes in studies involving large-scale flood inundation over longer durations (Yamazaki et al., 2018b).
Creating high-resolution flood inundation maps based on hydrodynamic modelling is computationally expensive
(Dottori et al., 2016) for a large domain like India. In addition, higher-resolution flood risk mapping that can be
used at the local scale for decision-making requires accurate terrain information and river cross-section datasets
that are not available. For instance, freely available digital elevation models (DEM) can be too coarse to resolve
the flood inundation and depth variability at a local scale (Cook & Merwade, 2009; Dey et al., 2022). The
uncertainties within hydrologic outputs can primarily arise due to inaccuracies in both input data and model
parameterization (Poulin et al., 2011). Inaccuracies in input meteorological data may stem from disparate sources,
leading to errors in spatial and temporal interpolation (Brown & Heuvelink, 2005). Similarly, model
parameterization errors, which involve assigning values to parameters governing diverse hydrological processes,
can emerge during the calibration process (Laiolo et al., 2015). Moreover, there are uncertainties originating from
utilizing long-term flood occurrence data to assess flood mapping capabilities. Our modelling framework that
considers the influence of reservoirs provides sub-basin scale flood inundation extent as our aim was to provide a
long-term assessment of flood extent in at the country scale. Additionally, downscaling of flood depths introduces
biases as coarse-scale information is translated to the local scale (He et al., 2021), which might have considerable
deviations from the actual observed flood extent. Given these limitations, our findings provide valuable
information based on the long-term record developed using model simulations that can be used for the regional
scale policy development for flood mitigation. Cloud cover during the summer monsoon, when most floods occur
in India (Nanditha et al., 2022), hinders the utility of satellite data for flood inundation mapping. We calibrated
and evaluated our H08-CaMa flood modelling framework using the observed flow, reservoir storage, and satellite-
based inundation. However, all these datasets available from the in-situ network or satellites are prone to errors
and uncertainty (Di Baldassarre & Montanari, 2009; Stephens et al., 2012; Teng et al., 2017). We used C-ratio as
an indicator to quantify the influence of dams on streamflow. However, C-ratio may not fully capture the
complexities and variations in the impacts of reservoir operations. Furthermore, in case of run-of-the-river (RoR)
dams, the C-ratio may over-estimate the downstream hydrological impacts. Therefore, C-ratio may not solely
capture the downstream hydrological effects resulting from dams. Nevertheless, it provides preliminary
information on the potential dam influence on the downstream flow.
India has implemented several flood risk mitigation measures at multiple government levels. The construction of
embankments along rivers is a common flood risk mitigation measure in India. These embankments help contain
the floodwaters within the river channels and protect nearby areas from inundation (NDMA, 2016). The CWC in
India operates a network of flood forecasting stations that collect real-time data on rainfall and water levels to
forecast floods and issue warnings to vulnerable communities. Notwithstanding the considerable investments and
flood-control measures, India has witnessed substantial mortality, human migration, and economic loss. Flood
mortality has increased mainly because of increased frequency, not necessarily due to increased flood intensity
(Hu et al., 2018). About 3% of the total geographical area of India is affected by floods every year that cause
damage to agriculture and infrastructure. The top ten floods that occurred during 1985-2015 caused the mortality
of more than 1000 people while more than 35 million people were displaced due to floods between 2000-2004
(Dartmouth Flood Observatory). The recent riverine floods in Uttarakhand and Kerala highlighted the growing
flood risk in India, which warrants the need for flood mitigation. The recent flood in August 2022 in Pakistan
caused an estimated loss of $30 billion. Both structural and non-structural measures are required for flood
mitigation (Nanditha & Mishra, 2021). Our risk assessment provides policy implications towards reducing
vulnerability to reduce the flood risk. Moreover, a sub-basin level ensemble forecast is needed to be used for early
flood warnings in the sub-basins with higher flood risk.
Based on our findings, the following conclusions can be made:
• The coupled hydrological and hydrodynamic modelling framework based on the H08-CaMa Flood model
was used to estimate the flood risk assessment in India. The hydrological modelling framework
performed well against the observed flow, reservoir storage, and satellite-based flood inundation. The
role of 51 major reservoirs was considered in flood risk assessment based on the long-term simulations
for the 1901-2020 period.
• The sub-basins in the Ganga and Brahmaputra river basins experienced the most significant flood extent
during the worst flood in 1901-2020. Similarly, the mean annual maximum flood extent is higher for the
sub-basins in the two major transboundary river basins (e.g., Ganga and Brahmaputra). The worst flood
affected different sub-basins on the two main flood-affected river basins in different years. Major floods
in the flood-prone sub-basins of the Ganga and Brahmaputra basins occur during the summer monsoon
season, especially during the August-September period.
• The sub-basins with a more prominent influence of dams based on the C-ratio were identified. Beas,
Brahmani, upper Satluj, Upper Godavari, Middle and Lower Krishna, and Vashishti sub-basins are
among the most influenced by the dams. Moreover, Beas, Brahmani, Ravi, and Lower Satluj are among
the most affected by floods and the presence of reservoirs.
• Flood risk is higher in the Ganga and Brahmaputra river basins compared to other parts of the country.
The higher flood risk in the two transboundary river basins can be attributed to higher vulnerability,
hazard probability, and exposure. Bhagirathi, Gandak, Kosi, lower Brahmaputra, and Ghaghra are India's
sub-basins with the highest flood risk.
**Data availability:** All the datasets used in this study can be obtained from the corresponding author.
**Competing interest:** Authors declare no competing interest.
**Author contributions:** VM designed the study. UV conducted the analysis and wrote the first draft. All the
authors contributed in the writing and discussion.
**Acknowledgement:** The work was supported by the Monsoon Mission, Ministry of Earth Sciences. The authors
acknowledge the data availability from India Meteorological Department (IMD) and India-WRIS. We
acknowledge the database availability from EM-DAT: http://www.emdat.be/, DFO:
http://floodobservatory.colorado.edu, population data from GHSL:
https://sedac.ciesin.columbia.edu/data/set/ghsl-population-built-up-estimates-degree-urban-smod, vulnerability
assessment data from DST: HYPERLINK
"https://dst.gov.in/sites/default/files/Full%20Report%20%281%29.pdf"https://dst.gov.in/sites/default/file
s/Full%20Report%20%281%29

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

821   9326/aa7250

822