# Peer review of "Flood risk assessment for Indian sub-continental river basins"

_Hydrology and Earth System Sciences, 2023_

## Author Response (AR1)

**Response to reviewers' comments**

**Anonymous Referee #1, 04 May 2023**

This study presents an important national-scale assessment of flood risk in India. The methodology is adequate and the manuscript is well written. I thus recommend its acceptance in HESS after minor review.

My only concern is that it is not clear throughout the text how reservoirs were considered in the study. In some parts of the methodology it is argued that the model is calibrated with dam storage data, and that the model considers the role of dams. Yet later it becomes more clear that actually the C-ratio was used, which is a proxy of storage effects; i.e. dams were actually not simulated by the model. Please clarify it across the text. Also, C-ratio is a proxy of storage effects and has some limitations. RoR dams can sometimes have high C-ratio but low impacts on downstream hydrology. Please add something about it to the discussion.

Thanks. We appreciate the constructive comments and feedback and agree that it is important to provide a clear and consistent explanation throughout the text. We have revised the manuscript to ensure a more coherent presentation on the methods to include reservoirs in our study.

We have revised the text related to C-ratio. We would like to highlight that the model considers the role of dams and C-ratio is not a proxy of storage effects, rather it is an indicator which is estimated to quantify the influence of dams on streamflow. However, we acknowledge that the C-ratio has its limitations and may not fully capture the complexities and variations in the impacts of reservoir operations. Specifically, we recognize that in the case of run-of-the-river (RoR) dams, despite having a high C-ratio, may have limited downstream hydrological impacts due to their operational characteristics. In the revised manuscript, we have added clarifications and included a discussion that highlights this point, emphasizing that the C-ratio alone may not provide a comprehensive assessment of the effects of dams on downstream hydrology. Therefore, we have revised the manuscript to provide a more accurate and thorough understanding of how reservoirs, including RoR dams, were considered in our study.

(L128-132)

*"We calibrated the combined model (H08 and CaMa-Flood) for India's eighteen major river basins for one gauge station, each considering the influence of 51 major dams. The gauge stations were selected in the farthest downstream of the river basin based on the availability of observed streamflow. The dam operations were simulated through the CaMa-Flood model and evaluated against the observed daily live reservoir storage data."*

(L394-399)

*"We used C-ratio as an indicator to quantify the influence of dams on streamflow. However, C-ratio may not fully capture the complexities and variations in the impacts of reservoir operations. Furthermore, in case of run-of-the-river (RoR) dams, the C-ratio may over-estimate the downstream hydrological impacts.*

*Therefore, relying solely on the C-ratio cannot furnish a thorough evaluation of the downstream hydrological effects resulting from dams. Nevertheless, C-ratio can provide preliminary information on the potential dam influence on the downstream flow."*

Minor comments:

Line 15-16 This sentence is too vague, please rephrase it: 'Sub-basins in the Ganga and Brahmaputra River basins witnessed the greatest flood extent during the worst flood in the observational record.'

Thanks. We have addressed the comment in the revised manuscript. (L15-16)

*"Sub-basins in the Ganga and Brahmaputra River basins witnessed substantial inundation extent during the worst flood in the observational record."*

L. 66 This reference on the number of dams is from 2000. Is this number updated?

Thanks. We have addressed the comment in the revised manuscript.

*"India has more than 5300 large dams regulating river flow (National Register of Large Dams (NRLD), 2019), affecting ecosystems, natural resources, and livelihoods (Acreman, 2000)."*

L. 100 GSW cannot estimate flooded vegetation and is usually limited for flood events because of cloud influence. Furthermore, your analysis is for 1901-2020, yet GSW covers only 1985-present. Please clarify the use of GSW in the text.

Thanks. We have addressed the comment in the revised manuscript. (L102-105)

*"We utilized the Global Surface Water (GSW) extent to estimate flood occurrences on a monthly timescale (Pekel et al., 2016). The simulated flood occurrences for the time period consistent with the availability of the GSW database (1985-2020) were employed to validate the model's performance in simulating flooded extents."*

L. 121 At this stage, it seems that you've simulated the dam impact on downstream hydrology, but later in paragraph 138-148 you mention that the dam potential impact on hydrology was calculated with the C-ratio. Please clarify it throughout the text.

Thanks. The comment has been addressed in the revised manuscript and we have clarified in the text.

(L130-131)

*"The dam operations were simulated through the CaMa-Flood model and evaluated against the observed daily live reservoir storage data."*

(L144-155)

*"We used C-ratio (Nilsson et al., 2005; Zajac et al., 2017) to assess the potential impact of dams along a river. The C-ratio is an identifier calculated as the ratio of upstream reservoirs' total maximum storage capacity to the mean annual discharge at a selected point along the river downstream (Nilsson et al., 2005; Zajac et al., 2017). We calculated the C-ratio at the outlets of each sub-basins that are influenced by the presence of dams. A C-ratio of less than 0.5 indicates that the sub-basin is minimally affected by the presence of dams. Further, to identify sub-basins susceptible to flood inundation resulting from dam operations, we multiplied the percentage of flooded area in each sub-basin by its corresponding C-ratio. This allowed us to identify the sub-basins that experience substantial flood inundation and are significantly impacted by the presence of reservoirs.  Finally, we estimated the exposed rail and road infrastructure affected by floods. The flooded area overlapped over the road and railway network to estimate the network length affected by floods in a sub-basin. We considered the flooded area of the observed worst flood. The subbasins with the highest rail and road infrastructure exposure to floods were identified."*

L. 212-213 To illustrate the fact that ' the worst flood in the same year did not affect all the sub-basins within a river basin', it would be very interesting to see a heat map of the drainage network colored according to the year of the worst flood in the century. It would be easy to make it based on the model outputs.

Thanks. We have addressed the comment in the revised manuscript.

[Figure]

**Figure S6. Heat map illustrating worst flood year in different streams**

L. 221 It would be important to present in the larger map the locations of the sub-basins (a to j)

Thanks. Done.

[Figure]

a) Ghaghara (03/09/1915)

b) Kosi (19/08/1914)

c) Bhagirathi and others
(01/10/1924)

d) Gandak and others (14/09/1948)

e) Upstream of gomti confluence
to muzaffarnagar (10/10/1924)

f) Gomti (14/09/1915)

g) Lower Sabarmati
(28/07/1927)

h) Upper Yamuna
(30/09/1924)

i) Ramganga
(20/09/1914)

j) Baitarani
(05/08/1943)

Flood water depth (m)

**Figure 3: Flood depth map for the observed worst flood for each sub-basins, highlighting the sub-basins with maximum flood inundated area (%) (a) Ghaghara – Ganga River basin (b) Kosi – Ganga River basin (c) Bhagirathi and others – Ganga River basin (d) Gandak and others – Ganga River basin (e) Upstream of Gomti confluence to Muzaffarnagar – Ganga River basin (f) Gomti – Ganga River basin (g) Lower Sabarmati – Sabarmati River basin (h) Upper Yamuna – Ganga River basin (i) Ramganga – Ganga River basin (j) Baitarani – Brahmani River basin**

L. 234-234 Are there any references (especially in grey literature) that could be used to support your findings of when the worst flood occurred in the different sub-basins?

Thanks. We have addressed the comment in the revised manuscript.

*"Our findings align with the information presented in various news articles and reports (Agarwal & Narain, 1991; Fredrick, 2017; Joshi, 2014; D. K. Mishra, 2015; A. Singh et al., 2021)."*

L. 279 Change to uppercase: 'C-ratio'

Thanks. Done.

L. 338 Typo: 'Hirabayashi et al. (2013)'

Thanks. Done.

L. 336 I missed in the discussion some comments about which mitigation measures are being undertaken by India at multiple government levels.

Thanks. We have revised the discussion. (L399-403)

*"India has implemented several flood risk mitigation measures at multiple government levels. The construction of embankments along rivers is a common flood risk mitigation measure in India. These embankments help contain the floodwaters within the river channels and protect nearby areas from inundation (NDMA, 2016). The CWC in India operates a network of flood forecasting stations that collect real-time data on rainfall and water levels to forecast floods and issue warnings to vulnerable communities."*

**Anonymous Referee #2, 08 Jun 2023**

The study uses a modeling approach aiming to assess the flood risk at the sub-basin scale in India and the impact of reservoirs on the flood risk. My recommendation is rejection due to several significant issues in the methodology that seem inadequate for addressing the posed questions. Further justification is also necessary for the conclusions made. Here are my main concerns:

**The understanding of flood risk seems to focus too heavily on the worst flood event in history. To understand flood risk, it requires examination of a large number of flood events over a range of conditions and incorporating uncertainties.**

Thanks. We appreciate insightful comments and suggestions that have been addressed in the revised manuscript. We have included more observed flood events of varying intensities in the revised manuscript. By incorporating multiple flood events, our revised analysis captures the variability in flood characteristics and associated uncertainty. In addition, our revised analysis examines the flood risk at the sub-basin scale in a robust manner and provides a comprehensive understanding of the potential impacts. We would like to highlight that there is a lack of observations at an appropriate spatial and temporal resolution that can be used for flood risk assessment in India at the sub-basin scale. This is probably the first study that attempts to reconstruct the flood risk at the sub-basin scale in India based on the last 120 years of data. We have considered the worst flood and also the annual maximum flood to estimate the flood-affected regions. Please see Fig. 6 in the revised manuscript for further details.

**I question the suitability of a large-scale model like H08-CaMaFlood for flood risk assessment, which typically requires higher-resolution models that can accurately capture local topography and features. Given the shown substantial bias in simulated flood occurrences, I am unconvinced of the model's efficacy in predicting flood water depth at the event scale. The downscaling approach, which scales simulated flood depth from a 0.1 degree to a 200 m resolution within CaMa-Flood, compounds this uncertainty. Ultimately, the rationale behind the choice of CaMa-Flood for localized flood risk assessment is unclear to me, as its resolution seems too coarse for the purpose.**

Thanks for the comment and critical insights. We are aware that flood risk at the local scale requires a high-resolution (sub-meter) scale of hydrological and hydraulic modeling. However, the aim of the current work is not to provide a flood-risk assessment for the entire country based on high-resolution modeling and data. This would be beyond the scope of the work to reconstruct flood maps at high resolution using the database for more than 120 years. In the revised manuscript, we have clearly mentioned the scope of the current work and its limitations.

We appreciate the comments and suggestions on the use of the H08-CaMa-Flood model for flood risk assessment. While it is true that flood risk assessment requires higher-resolution models to accurately capture local topography and features, it is also important to consider the specific context and objectives of the assessment.

The suitability of a large-scale model like H08-CaMa-Flood for flood risk assessment can depend on several factors. One such factor is the scale of the study area. In our case, the assessment is focused on a broader region where simulating high-resolution inundation dynamics for the entire area may not be

feasible due to computational limitations and high computation time. In such situations, a large-scale model like H08-CaMa-Flood can provide a valuable overview and identify areas of higher flood risk that can be further investigated using more detailed methods, if necessary.

While local-scale models may be preferred for predicting flood water depth at the event scale (Bates et al., 2010), the aim of our study is not to precisely estimate water depths but rather to identify areas at risk and assist in prioritizing resources for mitigation efforts. In such cases, a large-scale model can still provide valuable insights by indicating areas that are more susceptible to flooding, allowing decision-makers to allocate resources and develop appropriate strategies. Therefore, we believe that our study will provide crucial insights into large-scale drivers and patterns of flood risk to develop more informed adaptation and mitigation measures. We have revised the manuscript to clarify these issues and highlighted the novel contribution of our study, along with potential limitations. While H08-CaMa-Flood exhibited some bias, its performance against other available models cannot be ignored (Hirabayashi et al., 2013; Yamazaki et al., 2011). Therefore, our modeling framework for the sub-basin scale flood risk assessment in India can provide important insights that can help in flood mitigation.

For the downscaling approach within the CaMa-Flood model, it is true that there are uncertainties associated with the process of scaling simulated flood depth from a larger grid resolution to a finer resolution (Yamazaki et al., 2017). However, the downscaling approach is often utilized to make the outputs of large-scale models more applicable at local scales. We have compared the flood risk maps at the original model resolution against the downscaled maps/observational datasets and discussed the limitation in the revised manuscript.

**The authors' claim of an acceptable model skill is unconvincing to me. For river flooding, they set a NSE threshold of 0.5, which is questionable since a score of 0.6 is generally considered the minimum for model adequacy. Even then, some stations fail to meet this lowered threshold. There is also a lack of flood-relevant metrics, such as bias in peak discharge of flood events.**

While different thresholds for NSE (Nash-Sutcliffe Efficiency) for satisfactory model performance are available in the published literature, it's important to note that there is no universally agreed-upon threshold for model adequacy in streamflow prediction. Regarding the note that an NSE threshold of 0.5 for river flooding is questionable, although it is arguable, that this threshold is commonly used and accepted in the field, especially for daily streamflow prediction (Dakhlalla & Parajuli, 2019; Leta et al., 2018). However, it's true that higher NSE values are generally desired for more accurate predictions.

The mention of some stations failing to meet even the lowered threshold is indeed a valid concern. We have improved the model calibration for these stations, re-evaluated the model parameters and input data, and conducted a comprehensive analysis to improve the model's performance in the revised manuscript. We thank the reviewer for highlighting the need for flood-relevant metrics, specifically the bias in peak discharge of flood events. In the revised manuscript, we have included an additional plot showing the bias and timing error in peak discharge for each river basin.

[Figure]

**Figure 1: Calibration and evaluation of the combined model for daily river flow and reservoir storage at gauge stations and daily live storage of reservoirs**

[Figure]

**Figure S02: a) Bias (percentage) and b) Timing error in the simulation of yearly maximum streamflow events**

**I would suggest evaluating the worst flood event selected as well. Concerning flood inundation modeling, it would be beneficial if flood extent data were used to evaluate the model's skill. With respect to flood occurrences, I noted previously that the bias seems significant even before the application of downscaling. Despite these evident issues, no discussions on the uncertainties present in this study are included. This omission casts further doubt on the reliability of the results and necessitates a comprehensive review of the methodology.**

Thanks. In terms of flood inundation modeling, incorporating flood extent data for evaluating the model's skill is an excellent suggestion. Flood extent data can offer a more direct measure of the model's performance in simulating flood dynamics and spatial patterns. Including such an evaluation would strengthen the study's methodology and provide a clearer understanding of the model's capabilities. We have re-examined the model's ability in simulating the flood extent as well for some selected flood events. In addition, we have discussed the causes of the bias and highlighted potential sources (input data, model parameterization) of uncertainty. Further, the omission of discussions on uncertainties is indeed a noteworthy point. We have addressed the suggestions related to limitations and uncertainty in flood risk assessment in the revised manuscript and provided a separate section in the discussion of the manuscript. (L377-390)

*"The uncertainties within hydrologic outputs can primarily arise due to inaccuracies in both input data and model parameterization (Poulin et al., 2011). Inaccuracies in input meteorological data may stem from disparate sources, leading to errors in spatial and temporal interpolation (Brown & Heuvelink, 2005). Similarly, model parameterization errors, which involve assigning values to parameters governing diverse hydrological processes, can emerge during the calibration process (Laiolo et al., 2015). Moreover, concerns surround the uncertainties originating from utilizing long-term flood occurrence data to assess flood mapping capabilities. This data type provides generalized statistics across multiple flood events, masking the nuances of individual events and potentially leading to biased assessments. Additionally, the act of downscaling flood depths introduces biases as coarse-scale information is translated to the local scale (He et al., 2021). Consequently, these downscaled estimates might significantly deviate from actual conditions, thereby causing both overestimation and underestimation of flood risks. Given these limitations, our findings provide valuable information based on the long-term record developed using model simulations that can be used for the regional scale policy development for flood mitigation. Cloud cover during the summer monsoon, when most floods occur in India (Nanditha et al., 2022), hinders the utility of satellite data for flood inundation mapping."*

[Figure]

**Figure 3: Simulated flood extent compared with Sentinel-1 SAR and MODIS satellite-based observed flood extent for the 2016 flood event in the Brahmaputra river**

[Figure]

**Figure S03: Simulated flood extent compared with Sentinel-1 SAR and MODIS satellite-based observed flood extent for the 2016 flood event in the Ganga river basin**

[Figure]

**Figure S04: Simulated flood extent compared with Sentinel-1 SAR and MODIS satellite-based observed flood extent for the 2019 flood event in the Brahmaputra river basin**

**The use of the C-ratio to assess the role of reservoir operations in flood risk is confusing. The C-ratio, defined as the ratio of a reservoir's total maximum storage capacity to the mean annual discharge at the sub-basin outlet, is essentially a constant that doesn't account for variability in reservoir outflow resulting from operations serving different objectives. The mean annual discharge also seems irrelevant when examining a record flood event at a much shorter timescale. Consequently, I find the results based on C-ratio to be lacking in significance. Certain fundamental details that could aid in interpreting the results are missing, such as a clear definition of how a flood event is defined.**

Thanks. We appreciate the suggestion and acknowledge that the C-ratio does not capture the full complexity of reservoir operations and their impact on flood risk. However, we would like to highlight that incorporating the reservoir operations related complexities is challenging due to lack of observational datasets related to reservoir operations. Therefore, our aim was to provide an overview of the sub-basins that have high flood risks and affected substantially by the reservoir operations. While the C-ratio may have limitations in assessing the influence of reservoir operations during extreme flood events at shorter timescales, it can still provide a useful measure of the potential storage capacity available in a reservoir relative to the average discharge over a longer term. We have provided more details on the utility of C-ratio in the revised manuscript. (L144-155)

*"We used C-ratio (Nilsson et al., 2005; Zajac et al., 2017) to assess the potential impact of dams to estimate the potential dam effect along a river. The C-ratio is an identifier calculated as the ratio of a upstream reservoirs' reservoir's total maximum storage capacity to the mean annual discharge at a selected point along the river downstream (Nilsson et al., 2005; Zajac et al., 2017). We calculated the C-ratio at the outlets of each sub-basins that are influenced by the presence of dams. A C-ratio of less than 0.5 indicates that the sub-basin is minimally affected by the presence of dams. A lower (less than 0.5) C-ratio indicates that the sub-basin is not considerably affected by the presence of dams. Further, to identify sub-basins susceptible to flood inundation resulting from dam operations, we multiplied the percentage of flooded area in each sub-basin by its corresponding C-ratio. Further, we multiplied the percentage flooded area of each sub-basin with their corresponding C-ratio, which was used to identify the sub-basins that experience considerable flood inundation and are affected by the presence of reservoirs. This allowed us to identify the sub-basins that experience substantial flood inundation and are significantly impacted by the presence of reservoirs. The identified sub-basins are prone to flooding due to dam operations. Finally, we estimated the exposed rail and road infrastructure affected by floods. The flooded area overlapped over the road and railway network to estimate the network length affected by floods in a sub-basin. We considered the flooded area of the observed worst flood. The subbasins with the highest rail and road infrastructure exposure to floods were identified."*

**References**

Bates, P. D., Horritt, M. S., & Fewtrell, T. J. (2010). A simple inertial formulation of the shallow water equations for efficient two-dimensional flood inundation modelling. *Journal of Hydrology*, *387*(1–2), 33–45. https://doi.org/10.1016/J.JHYDROL.2010.03.027

Dakhlalla, A. O., & Parajuli, P. B. (2019). Assessing model parameters sensitivity and uncertainty of streamflow, sediment, and nutrient transport using SWAT. *Information Processing in Agriculture*, *6*(1), 61–72. https://doi.org/10.1016/J.INPA.2018.08.007

Hirabayashi, Y., Mahendran, R., Koirala, S., Konoshima, L., Yamazaki, D., Watanabe, S., Kim, H., & Kanae, S. (2013). Global flood risk under climate change. *Nature Climate Change*, *3*(9), 816–821. https://doi.org/10.1038/nclimate1911

Joshi, V. (2014, September 14). Have we learnt from past floods? Clearly not! *Hindustan Times (Lucknow)*. https://www.pressreader.com/india/hindustan-times-lucknow/20140914/281646778342401

Leta, O. T., El-Kadi, A. I., & Dulai, H. (2018). Impact of Climate Change on Daily Streamflow and Its Extreme Values in Pacific Island Watersheds. *Sustainability 2018, Vol. 10, Page 2057*, *10*(6), 2057. https://doi.org/10.3390/SU10062057

Yamazaki, D., Ikeshima, D., Tawatari, R., Yamaguchi, T., O'Loughlin, F., Neal, J. C., Sampson, C. C., Kanae, S., & Bates, P. D. (2017). A high-accuracy map of global terrain elevations. *Geophysical Research Letters*, *44*(11), 5844–5853. https://doi.org/10.1002/2017GL072874

Yamazaki, D., Kanae, S., Kim, H., & Oki, T. (2011). A physically based description of floodplain inundation dynamics in a global river routing model. *Water Resources Research*, *47*(4), 4501. https://doi.org/10.1029/2010WR009726

---

## Referee Report (RR1)

1. The large-scale H08-CaMaFlood model used by the authors successfully (NSE>0.6) reproduced hydrographs of daily flow over a long-term period in almost all river basins under consideration. For me, this result is quite unexpected, since, as a rule, global hydrological models poorly reproduce the seasonal variation of river flow, even with flow averaging over larger time intervals than a day (Hattermann et al., 2017; Krysanova et al., 2018). Specifically, H08 model, as far as I can judge from publications, has not yet given such good results for river basins located in monsoon climates (see, for example, Yoshida et al., 2022). The result obtained by the authors is important because it can expand our understanding of the effectiveness of global hydrological models at the scale of river basins. Taking this into account, I would like to see a more detailed description of the methods for setting the model's parameters, its calibration and verification, and other important, from the authors' point of view, details that made it possible to achieve this result.

2. The presented results for reproducing high flow and flood inundation (Fig. 2, 3, S2-S4) do not convince one of the possibility of using them to answer the research question formulated in the article: "How does the flood risk vary at the sub-basin scale in India for the observed worst floods that occurred during the 1901-2020 period?". This is not surprising since flood risk assessments require the use of rainfall-runoff and hydrodynamics models of much greater spatial resolution, which are able to take into account local runoff formation mechanisms, local topography, etc. The authors are well aware of the limitations of the model used and clearly articulate this in the Discussion section. At the same time, the authors pay excessive, in my opinion, attention to the analysis of specific catastrophic floods and the comparison of their simulated and observed characteristics, including using satellite-based flood extent data. Given such high uncertainty in modeling results, their agreement with observed data may be coincidental. I recommend shifting the focus of the article to the analysis of characteristics averaged over a long-term period (such an analysis is illustrated in Figures 6-9) by changing the research question as: "How does the flood risk vary at the sub-basin scale in India during the 1901-2020 period?"

3. The article must be formatted in accordance with the requirements of the journal.

Hattermann, F.F., et al. (2017) Cross-scale intercomparison of climate change impacts simulated by regional and global hydrological models in eleven large scale river basins. Climatic Change, 141 (3), 561–576. doi:10.1007/s10584-016-1829-4

Krysanova, V., Donnelly, C., Gelfan, A., Gerten, D., Arheimer, B., Hattermann, F., Kundzewicz, Z.W. (2018) How the performance of hydrological models relates to credibility of projections under climate change. Hydrological Sciences Journal, 63(5), 696-720 DOI: 10.1080/02626667.2018.1446214

Yoshida, T., Hanasaki, N., Nishina, K., Boulange, J., Okada, M., & Troch, P. A. (2022). Inference of parameters for a global hydrological model: Identifiability and predictive uncertainties of climatebased parameters. Water Resources Research, 58, e2021WR030660. https://doi.org/10.1029/2021WR030660

---

## Author Response (AR2)

Dear Editor,

We appreciate comments and suggestions from the reviewer/editor. We have carefully addressed the comments in the revised manuscript. Please see the details below.

Best regards
Vimal

**Response to the reviewer's comments**

1. The large-scale H08-CaMaFlood model used by the authors successfully (NSE>0.6) reproduced hydrographs of daily flow over a long-term period in almost all river basins under consideration. For me, this result is quite unexpected, since, as a rule, global hydrological models poorly reproduce the seasonal variation of river flow, even with flow averaging over larger time intervals than a day (Hattermann et al., 2017; Krysanova et al., 2018). Specifically, H08 model, as far as I can judge from publications, has not yet given such good results for river basins located in monsoon climates (see, for example, Yoshida et al., 2022). The result obtained by the authors is important because it can expand our understanding of the effectiveness of global hydrological models at the scale of river basins. Taking this into account, I would like to see a more detailed description of the methods for setting the model's parameters, its calibration and verification, and other important, from the authors' point of view, details that made it possible to achieve this result.

Thanks. We appreciate the comment. The model calibration is crucial for the performance of the any hydrological models. Most global scale hydrological models are often not calibrated due to lack of data and/or the effort that is needed. Moreover, our modeling setup has been well calibrated for streamflow and reservoir storage. Moreover, we have also evaluated the performance of the H08 model against the satellite-based ET and soil moisture (Kushwaha et al., 2021; Journal of Hydrology). Therefore, the model shows relatively a better performance compared to global scale studies.

We have added the following text in the revised manuscript (Lines 131-141):

*"Large-scale global hydrological models do not perfectly capture the observed trends and variations as these are often not well calibrated at river basin scale (Krysanova et al., 2018). The H08 model performs well when calibrated at the river basin scale rather than coarser domains such as climate zones (Chuphal & Mishra, 2023; Yoshida et al., 2022). Here, we manually calibrated the H08 model by adjusting four key parameters that considerably influence streamflow for each river basin, which include single-layer soil depth, gamma, bulk transfer coefficient, and tau (Hanasaki et al., 2008; Raghav & Eldho, 2023). A more detailed discussion about the calibration parameters of H08 are discussed in Dangar & Mishra (2021). Different sets of combinations of calibration parameters within a range were used to calibrate the H08 model. The employed sets of parameters for the 18 river basins in the Indian sub-continent are listed in Table S2. The calibrated parameters account for the effect of human interventions because the model calibration is performed against the observed streamflow rather than the naturalized streamflow (Duc Dang et al., 2020)."*

2. The presented results for reproducing high flow and flood inundation (Fig. 2, 3, S2-S4) do not convince one of the possibility of using them to answer the research question formulated in the article: "How does the flood risk vary at the sub-basin scale in India for the observed worst floods that occurred during the 1901-2020 period?". This is not surprising since flood risk assessments require the use of rainfall-runoff and hydrodynamics models of much greater spatial resolution, which are able to take into account local runoff formation mechanisms, local topography, etc. The authors are well aware of the limitations of the model used and clearly articulate this in the Discussion section. At the same time, the authors pay excessive, in my opinion, attention to the analysis of specific catastrophic floods and the comparison of their simulated and observed characteristics, including using satellite-based flood extent data. Given such high uncertainty in modeling results, their agreement with observed data may be coincidental. I recommend shifting the focus of the article to the analysis of characteristics averaged over a long-term period (such an analysis is illustrated in Figures 6-9) by changing the research question as: "How does the flood risk vary at the sub-basin scale in India during the 1901-2020 period?"

Thanks. We acknowledge the limitations of our current model, which have been highlighted in the discussion section of the revised manuscript. We also understand the need for more sophisticated spatial resolution in rainfall-runoff and hydrodynamics models for accurate flood risk assessments, considering local runoff formation mechanisms and topography. However, the choice of hydrological/hydrodynamic models depends on the objective of the study. Our main aim is to identify the sub-basins in India that have high flood risk based on the long-term (1901-2020) observational record, for which, we feel that our hydrological modeling framework has satisfactory performance. Moreover, the smallest sub-basin in our study is larger than 5000 $km^2$ and at this scale the model performance to examine the flood risk can be considered reasonable. We are aware of the high resolution (meter or sub-meter) scale hydrological/hydrodynamic models that need comprehensive data inputs (elevation, cross sections etc) and can be applied at sub-daily time scale to examine the impacts of flooding at a local scale (within a urban area). Since we provide regional scale assessment, the future work can use the high-resolution models to estimate flood risks within sub-basins at the selected stretches. As per your suggestion, we have revised the question from "How does the flood risk vary at the sub-basin scale in India for the observed worst floods that occurred during the 1901-2020 period?" to "How does the flood risk vary at the sub-basin scale in India during the 1901-2020 period?".

We have added the following text in the revised manuscript (Lines 382-402):

*"While mapping the flood risk at appropriate spatial resolution is complex and challenging, it is vital for disaster risk reduction. Flood inundation mapping that provides the spatial extent of flooding is crucial as the first responders use it during a flood emergency (Apel et al., 2009). There are several approaches to mapping flood inundation (Teng et al., 2017). Various hydrological models have been employed for conducting flood risk assessments at a global scale (Dottori et al., 2018; Gu et al., 2020; Tabari et al., 2021). For instance, Dottori et al. (2018) used the H08 model combined with CaMa-Flood model to estimate losses resulting from river flooding at the country level. Additionally, the LISFLOOD model (van der Knijff et al., 2010) at 5 km spatial resolution was used to estimate the river flood risk in Europe (Alfieri et al., 2018). Flood risk assessment at relatively larger scales are conducted using the coarse resolution land surface hydrological models. The objective of these large scale flood risk assessment is to identify regions that are flood-prone (Dottori et al., 2018; Gu et al., 2020; Tabari et al., 2021). On the other hand, high resolution flood inundation mapping is needed to understand the local flood risk and damage caused to particular infrastructure. For the analysis of flood inundation during a particular floodat a local scale, high-resolution models such as HEC-RAS and Mike FLOOD can be employed (Khalaj et al., 2021; Nguyen et al., 2016). High resolution flood risk mapping requires comprehensive information of high-resolution topography, cross-sections of channels,*

*and data associated to structural measures of flood protection. However, the smallest subbasin considered in our study has more than 5000 km² area (Fig S7), while most subbasins have area between 10,000 and 50,000 km², with Lower Yamuna being the largest subbasin, with an area of 124,867.19 km². Therefore, the performance of our modelling framework against the satellite and other observations can be considered satisfactory to provide a sub-basin scale flood risk assessment. Moreover, we used hydrodynamic modelling to develop long-term flood inundation maps for the Indian sub-basins. The long-term data (1901-2020) provides us a record of several floods, which can help in robust estimates of flood risk in different sub-basins."*

3. The article must be formatted in accordance with the requirements of the journal.

Thank you. We have formatted the article according to the journal requirements.

Hattermann, F.F., et al. (2017) Cross-scale intercomparison of climate change impacts simulated by regional and global hydrological models in eleven large scale river basins. Climatic Change, 141 (3), 561–576. doi:10.1007/s10584-016-1829-4
Krysanova, V., Donnelly, C., Gelfan, A., Gerten, D., Arheimer, B., Hattermann, F., Kundzewicz, Z.W. (2018) How the performance of hydrological models relates to credibility of projections under climate change. Hydrological Sciences Journal, 63(5), 696-720 DOI: 10.1080/02626667.2018.1446214
Yoshida, T., Hanasaki, N., Nishina, K., Boulange, J., Okada, M., & Troch, P. A. (2022). Inference of parameters for a global hydrological model: Identifiability and predictive uncertainties of climatebased parameters. Water Resources Research, 58, e2021WR030660. https://doi.org/10.1029/2021WR030660

Thanks for the suggestions. We have cited these in the revised manuscript.